

# RAVAN: Multi-Head Low-Rank Adaptation for Federated Fine-Tuning

**Arian Raje**[*]    **Baris Askin**    **Divyansh Jhunjhunwala**    **Gauri Joshi**
Department of Electrical and Computer Engineering
Carnegie Mellon University

## Abstract

Large language models (LLMs) have not yet effectively leveraged the vast amounts of edge-device data, and federated learning (FL) offers a promising paradigm to collaboratively fine-tune LLMs without transferring private edge data to the cloud. To operate within the computation and communication constraints of edge devices, recent literature on federated fine-tuning of LLMs proposes the use of low-rank adaptation (LoRA) and similar parameter-efficient methods. However, LoRA-based methods suffer from accuracy degradation in FL settings, primarily because of data and computational heterogeneity across clients. We propose RAVAN, an adaptive multi-head LoRA method that balances parameter efficiency and model expressivity by reparameterizing the weight updates as the sum of multiple LoRA heads $s_i\mathbf{B}_i\mathbf{H}_i\mathbf{A}_i$ in which only the core matrices $\mathbf{H}_i$ and their lightweight scaling factors $s_i$ are trained. These trainable scaling factors let the optimization focus on the most useful heads, recovering a higher-rank approximation of the full update without increasing the number of communicated parameters since clients upload $s_i\mathbf{H}_i$ directly. Experiments on vision and language benchmarks show that RAVAN improves test accuracy by 2–8% over prior parameter-efficient baselines, making it a robust and scalable solution for federated fine-tuning of LLMs.

## 1 Introduction

In recent years, the amount of data available on edge devices has increased exponentially, opening the doors for applications that perform machine learning (ML) at the edge. One such paradigm is federated learning (FL), a model training regime where edge devices, or "clients", collaboratively train a model without sharing their local data with a central server [27]. FL training offers a way to perform large-scale ML on a distributed network of clients by utilizing on-device data while reducing potential privacy risks. The primary challenge in FL is to design methods that are robust in the presence of both data heterogeneity [24, 20, 23, 1]—variations in clients' local data distributions—and computational heterogeneity [10, 8, 30]—differences in clients' computing capacities.

Recent literature has begun exploring the integration of large language models (LLMs) into FL frameworks, driven by the surge in LLM-based edge applications and the resultant need to leverage on-device data for training [40, 41]. Unfortunately, naively training LLMs in FL settings is intractable as a result of the memory constraints of edge devices and communication constraints of wireless networks. As a consequence, these works have examined the impact of parameter-efficient fine-tuning (PEFT) for LLMs in federated settings [17, 41]. These methods reduce the computational load of fine-tuning pretrained LLMs by scaling down the number of trainable parameters. A particularly

---

[*]Corresponding Author. `araje@andrew.cmu.edu`

39th Conference on Neural Information Processing Systems (NeurIPS 2025).

important PEFT method in FL is low-rank adaptation (LoRA) [16], where the update $\Delta\mathbf{W}$ is re-parameterized as $\mathbf{BA}$, the product of two low-rank matrices $\mathbf{B}$ and $\mathbf{A}$. The original pretrained model parameters are frozen throughout fine-tuning, and only the LoRA $\mathbf{B}$ and $\mathbf{A}$ parameters ever receive gradient updates, resulting in fine-tuning that is vastly more parameter-efficient than full-parameter fine-tuning. LoRA-based methods are a promising alternative to full-parameter fine-tuning in FL since clients only have to train and communicate the LoRA parameters, simultaneously addressing computation and communication bottlenecks.

However, LoRA-based methods are highly affected by client data heterogeneity (see Table 1) because restricting updates to a low-rank subspace deprives the model of the capacity needed to fit the diverse directions introduced by heterogeneous data. Moreover, directly extending LoRA to FL, as done in FedIT [41], leads to an inexactness problem during aggregation. Since $\mathbf{BA}$ is a proxy for the true model update

Table 1: Accuracy comparison on CIFAR-100 [22], non-I.I.D. clients (Dirichlet $\alpha = 0.3$)

| Method | I.I.D. | Non-I.I.D. |
|---|---|---|
| Full-FT | 89.78 | 85.17 |
| FedIT | 81.75 | 68.15 |
| FedEx-LoRA | 77.82 | 66.98 |
| FFA-LoRA | 78.17 | 59.89 |

$\Delta\mathbf{W}$, averaging the $\mathbf{B}$ and $\mathbf{A}$ parameters separately would be inconsistent with the true model update:

$$\left(\frac{1}{|\mathcal{C}^{(t)}|} \sum_{c \in \mathcal{C}^{(t)}} \mathbf{B}_c^{(t)}\right) \left(\frac{1}{|\mathcal{C}^{(t)}|} \sum_{c \in |\mathcal{C}^{(t)}|} \mathbf{A}_c^{(t)}\right) \neq \frac{1}{|\mathcal{C}^{(t)}|} \sum_{c \in |\mathcal{C}^{(t)}|} \mathbf{B}_c^{(t)} \mathbf{A}_c^{(t)} \tag{1}$$

where $\mathcal{C}^{(t)}$ is the selected client set at round $t$. Previous works that seek to address this exact aggregation issue suffer from accuracy loss and poor scalability in practical FL settings due to data and computational heterogeneity. FFA-LoRA [34] manages exact updates by freezing the $\mathbf{A}$ parameter at initialization but reduces the model expressivity relative to vanilla LoRA. FedEx-LoRA [32] adds the inexact residual $\frac{1}{|\mathcal{C}^{(t)}|} \sum_{c \in \mathcal{C}^{(t)}} \mathbf{B}_c^{(t)} \mathbf{A}_c^{(t)} - \left(\frac{1}{|\mathcal{C}^{(t)}|} \sum_{c \in \mathcal{C}^{(t)}} \mathbf{B}_c^{(t)}\right)\left(\frac{1}{|\mathcal{C}^{(t)}|} \sum_{c \in \mathcal{C}^{(t)}} \mathbf{A}_c^{(t)}\right)$ to the original pretrained weights $\mathbf{W}$ to get an exact update every round. However, the method substantially increases the communication cost of fine-tuning since the updated model weights also have to be communicated every round. Critically, these LoRA-based methods can afford only small ranks within a limited parameter budget. When the true update is high-rank, this approximation disregards most of the update's variance and limits accuracy. Fed-SB [33], motivated by the update approximation from LoRA-XS [5], introduces a third LoRA parameter between the standard $\mathbf{B}$ and $\mathbf{A}$ parameters and only fine-tunes this additional parameter. However, it necessitates an initial round of full-parameter fine-tuning to initialize $\mathbf{B}$ and $\mathbf{A}$, which is prohibitively expensive in FL. Furthermore, the initialization may become stale as training progresses due to data heterogeneity and partial participation.

Computational heterogeneity is an additional scalability challenge in practical FL systems. Clients may vary significantly in their hardware resources and computational capabilities, making it difficult for all clients to fine-tune models at the same scale and speed. The methods described above do not allow for LoRA parameters of different sizes across clients. HetLoRA [9] and FlexLoRA [4] allow clients to train varying-rank LoRA parameters, but the methods struggle in the presence of data heterogeneity and do not ensure exact aggregation. In this vein, our goal is to design a method for FL that 1) performs efficient computation and communication throughout the training procedure, 2) remains robust in the presence of heterogeneity, and 3) retains the property of exact aggregation.

To this end, we propose RAVAN[2], an adaptive multi-head LoRA method that sharply reduces the number of trainable parameters while maintaining accuracy in the presence of data and computational heterogeneity. We take inspiration from multi-head approaches such as HydraLoRA [35]; however, when naively ported to federated settings, those methods fail to guarantee exact aggregation and cannot raise the effective rank of the updates under a fixed parameter budget. Our design meets both requirements. RAVAN re-parameterizes each weight update $\Delta\mathbf{W}$ as a weighted sum of low-rank heads $s_i\mathbf{B}_i\mathbf{H}_i\mathbf{A}_i$, where the bases $\mathbf{B}_i$ and $\mathbf{A}_i$ are frozen at initialization and only the $\mathbf{H}_i$ parameters and lightweight scaling parameters $s_i$ are trained. We choose $\mathbf{B}_i$ and $\mathbf{A}_i$ with mutually orthogonal column and row spaces, respectively, and thus the heads combine to achieve a higher effective rank without exceeding the original LoRA parameter budget. When clients differ in resources, the most constrained devices can fine-tune only a subset of heads and leave the rest frozen. Uploading the products $s_i\mathbf{H}_i$ preserves exact aggregation and the method incurs no extra communication cost. Extending prior efforts, RAVAN introduces an integrated framework that maintains parameter-efficient computation

---

[2]RAVAN derives its name from the mythical 10-headed villain from the Hindu epic, *Ramayana*

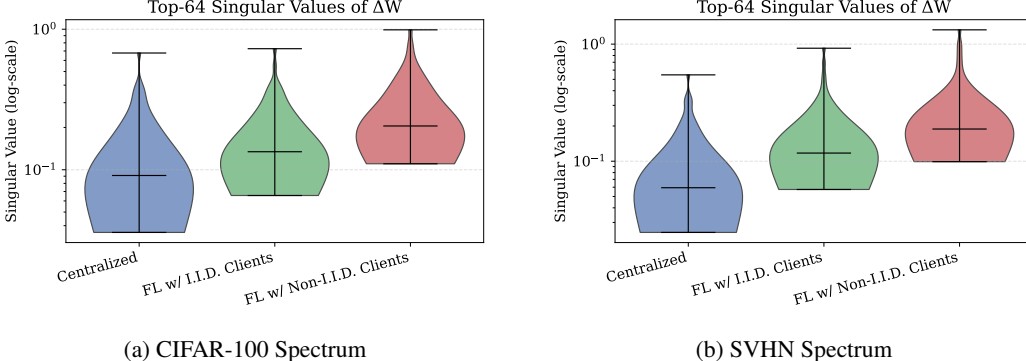

| (a) CIFAR-100 Spectrum | (b) SVHN Spectrum |

Figure 1: Singular value spectra of the weight updates $\Delta \mathbf{W}$ for CIFAR-100 and SVHN [29] in three different training regimes. We display only the 64 largest values (hence the truncated plots). Moving from centralized learning $\rightarrow$ FL (I.I.D. clients) $\rightarrow$ FL (non-I.I.D. clients), the median shifts up and the distribution becomes broader, meaning a larger fraction of singular values remains near the higher end of the spectrum. The effective rank is, therefore, higher in the federated, non-I.I.D. setting.

and communication and demonstrates robustness across diverse data and computational heterogeneity in federated settings. Across all benchmarks, RAVAN outperforms related federated PEFT methods in both I.I.D. and non-I.I.D. settings, demonstrating its strength in the presence of heterogeneity and client diversity.

## 2  Problem Setup and Motivation

LoRA is a PEFT method that reparameterizes the weight updates to reduce the number of trainable parameters. It contends that the full-parameter weight update $\Delta \mathbf{W}_{\text{full}}$ can be approximated as follows:

$$\underbrace{\Delta \mathbf{W}_{\text{full}}}_{\text{Weight Update}} \approx \underbrace{\mathbf{BA}}_{\text{LoRA Parameters}} \tag{2}$$

For notational ease, we write each weight matrix $\mathbf{W}$ and its corresponding update $\Delta \mathbf{W}_{\text{full}}$ as a square matrix in $\mathbb{R}^{d \times d}$, but all derivations extend directly to the general rectangular case in which $\mathbf{W} \in \mathbb{R}^{m \times n}$. The method contends that $\Delta \mathbf{W}_{\text{full}}$ exists in a low-rank subspace and can therefore be represented as the product of two low-rank matrices. In this context, the low-rank matrices, referred to as the LoRA parameters, have dimensions $\mathbf{B} \in \mathbb{R}^{d \times r}$ and $\mathbf{A} \in \mathbb{R}^{r \times d}$. To perform LoRA fine-tuning, the pretrained model weights $\mathbf{W}$ are frozen throughout training and only $\mathbf{B}$ and $\mathbf{A}$ receive gradient updates. Since $r \ll d$, the number of trainable parameters decreases from $d^2$ to $2rd$.

**Importance of Higher-Rank Update Approximation.**  A key limitation of LoRA is that constraining the approximation of the update $\Delta \mathbf{W}_{\text{full}}$ to the low-rank subspace spanned by $\mathbf{BA}$ can limit its expressive capacity. When the rank $r$ is set too low, the approximation of $\Delta \mathbf{W}_{\text{full}}$ may fail to capture the full complexity and variation present in the full-rank gradient updates. This limitation is amplified when we perform fine-tuning in a federated setting, as we observe in Figure 1 which shows the spectra of singular values of the *full-parameter* weight updates, $\Delta \mathbf{W}_{\text{full}}$, in three different training regimes (centralized learning, FL with I.I.D. clients, and FL with non-I.I.D. clients). The model is trained to a target accuracy, and the weight update is decomposed using singular value decomposition (SVD). Figure 1 demonstrates that the "effective rank" of the weight updates is larger in the federated non-I.I.D. setting. Intuitively, the greater the diversity among client updates, the more the spectral mass is spread across singular vectors, thereby increasing the effective rank. This suggests that low-rank approximations of the weight updates discard more information and fail to capture many of the significant intrinsic dimensions of the true updates.

**Improving the Effective Rank and Expressivity of LoRA.**  A naive way to capture the richer spectrum of weight updates is to raise the LoRA rank $r$, but that linearly increases the number of trainable parameters. To better approximate the higher-rank update, as proposed in LoRA-XS, we

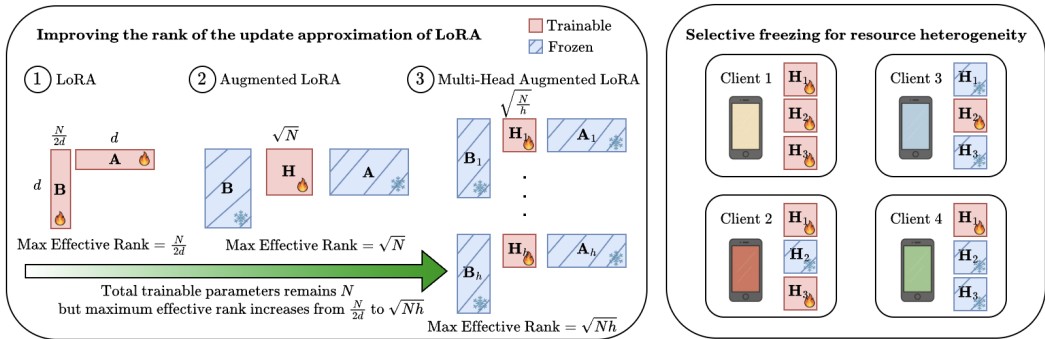

Figure 2: **Left**: Within the same parameter count, the effective rank of the LoRA parameters increases when using an augmented third parameter and multiple heads. **Right**: Clients with various computational constraints can freeze certain heads to reduce memory consumption.

can augment the traditional LoRA approximation with an additional parameter, $\mathbf{H}$, as follows:

$$\underbrace{\Delta \mathbf{W}_{\text{full}}}_{\text{Weight Update}} \approx \underbrace{\mathbf{BHA}}_{\text{LoRA Parameters}} \tag{3}$$

where $\mathbf{B}$ and $\mathbf{A}$ remain frozen and only $\mathbf{H}$ is trained. Suppose we have a trainable parameter budget of $N$ parameters. In the case of vanilla LoRA, $N = 2rd$ and $r = \Theta(1)$. With the augmented version of LoRA that has parameters $\mathbf{BHA}$ and frozen $\mathbf{B}$ and $\mathbf{A}$, we can instead use the much larger rank of $\sqrt{N} = \Theta(\sqrt{d})$. Additionally, if used in FL, this setup would avoid inexactness in the aggregation of the disparate client models since only the $\mathbf{H}$ parameter is averaged across clients.

**Multiple Heads for Further Rank Improvements.** We can further improve the effective rank of the update approximation by using multiple concurrent augmented LoRA heads. Again, suppose we have a trainable parameter budget of $N$ parameters. We use $h$ heads, where each head $i \in [1, \ldots, h]$ has the structure $\mathbf{B}_i \mathbf{H}_i \mathbf{A}_i$ and each $\mathbf{B}_i$ and $\mathbf{A}_i$ is frozen at initialization. With this reparameterization of the weight update, each head has rank $\sqrt{\frac{N}{h}}$. Using the sub-additivity of rank, we instead have:

$$\text{rank}\left(\mathbf{B}_i \mathbf{H}_i \mathbf{A}_i\right) = \sqrt{\frac{N}{h}} \ \forall i \in [1, \ldots, h] \implies \text{rank}\left(\sum_{i=1}^{h} \mathbf{B}_i \mathbf{H}_i \mathbf{A}_i\right) \leq h \cdot \sqrt{\frac{N}{h}} = \sqrt{Nh} \tag{4}$$

Within the same trainable parameter budget, by using $h$ heads, we can improve the rank expressivity of the augmented version of LoRA by a factor of $\sqrt{h}$. Furthermore, using heads of the form $\mathbf{B}_i \mathbf{H}_i \mathbf{A}_i$, where $\mathbf{B}_i$ and $\mathbf{A}_i$ are fixed, retains the property of exact aggregation because the following is true:

$$\frac{1}{|\mathcal{C}^{(t)}|} \sum_{c \in \mathcal{C}^{(t)}} \sum_{i=1}^{h} \mathbf{B}_i \mathbf{H}_{c,i}^{(t)} \mathbf{A}_i = \sum_{i=1}^{h} \mathbf{B}_i \left(\frac{1}{|\mathcal{C}^{(t)}|} \sum_{c \in \mathcal{C}^{(t)}} \mathbf{H}_{c,i}^{(t)}\right) \mathbf{A}_i \tag{5}$$

Note that methods like HydraLoRA and LoRAMoE [12] that use multiple vanilla LoRA heads of the form $\mathbf{B}_i \mathbf{A}_i$ (instead of the augmented $\mathbf{B}_i \mathbf{H}_i \mathbf{A}_i$ form) do not confer these same benefits of increased rank expressivity and exact aggregation. Suppose we have $N$ trainable parameters; each vanilla LoRA head would have dimensions $\mathbf{B}_i \in \mathbb{R}^{d \times \frac{N}{2dh}}$ and $\mathbf{A}_i \in \mathbb{R}^{\frac{N}{2dh} \times d}$. We would then have the same maximum effective rank as vanilla LoRA because of the following inequality:

$$\text{rank}\left(\mathbf{B}_i \mathbf{A}_i\right) = \frac{N}{2dh} \ \forall i \in [1, \ldots, h] \implies \text{rank}\left(\sum_{i=1}^{h} \mathbf{B}_i \mathbf{A}_i\right) \leq h \cdot \frac{N}{2dh} = \frac{N}{2d} \tag{6}$$

Since the number of trainable parameters $N = \Theta(d)$ (recall that $N = 2dr$ for standard LoRA), this rank is $N/2d = \Theta(1)$, which is much smaller than the rank $\sqrt{Nh} = \Theta(\sqrt{dh})$ achieved by multiple augmented LoRA heads.

**Addressing Computational Heterogeneity.** In a realistic resource-heterogeneous federation, clients may have vastly different computational capacities. A PEFT scheme that forces every device to train the same size LoRA parameters will exclude the weakest clients or throttle the strongest. In the FL setting specifically, using multiple heads has the additional benefit of providing a way to manage computational heterogeneity. Clients with more significant resource limitations can freeze subsets of the heads and only fine-tune the remaining heads (Figure 2, **Right**). This further reduces the memory requirements of local fine-tuning and prevents clients from having to drop out of the FL procedure due to stricter resource constraints. This partial freezing scheme still avoids inexactness in the aggregate updates, unlike previous heterogeneous-rank works in FL.

## 3 Proposed Method

In this section, we present RAVAN, a method that uses multiple augmented LoRA heads to perform efficient LLM fine-tuning in the presence of data and computational heterogeneity. For pretrained weights $\mathbf{W}$ in the model $\mathcal{M}$, the forward pass is replaced by the following:

$$\underbrace{\left( \mathbf{W} + \sum_{i=1}^{h} s_i \mathbf{B}_i \mathbf{H}_i \mathbf{A}_i \right) \boldsymbol{x}}_{\text{RAVAN Forward Pass}} \quad (7)$$

The pretrained $\mathbf{W}$ along with each $\mathbf{B}_i$ and $\mathbf{A}_i$ are frozen at the start of training. As a consequence, only the $\mathbf{H}_i$ parameters and the lightweight scaling factors $s_i$ are updated and communicated throughout the FL training procedure. The pseudocode of the proposed method RAVAN is given in Algorithm 1, and the following sections highlight key components of our framework. Specifically, we examine the importance of initialization in improving the update approximation (Section 3.1). We additionally analyze strategies for per-client head subset selection and aggregation for computational heterogeneity (Section 3.2). Together, these design choices let RAVAN match the communication cost of vanilla LoRA, while delivering higher-rank, resource-aware updates that preserve exact aggregation.

---

**Algorithm 1** RAVAN

**Require:** Clients $\mathcal{C}$, Model $\mathcal{M}$, Rounds $T$, Local Steps $S$, LR $\ell$, Rank $r$, Heads $h$
1: **Initialization**:
2: $\text{INIT}(\mathbf{B}_i, \mathbf{H}_i^{(0)}, \mathbf{A}_i), i = [1, \ldots, h]$ for $\mathcal{M}$
3: Freeze original model parameters, $\mathbf{B}_i, \mathbf{A}_i$
4: **Model Training**:
5: **for** $t = 1$ **to** $T$ **do**
6:     Reset $s_i^{(t)} \leftarrow 1, i \in [1, \ldots, h]$
7:     Select active client subset $\mathcal{C}^{(t)}$
8:     Broadcast $\{\mathbf{H}_i^{(t)}\}_{i=1}^{h}$ to $\mathcal{C}^{(t)}$
9:     **for all** $c \in \mathcal{C}^{(t)}$ **in parallel do**
10:         $\mathcal{H}_c^{(t)} \leftarrow \text{SELECTHEADS}(c, t)$
11:         **for** $\tau = 1$ **to** $S$ **do**
12:             Update $s_i^{(t)}, \mathbf{H}_i^{(t)}$ for $i \in \mathcal{H}_c^{(t)}$
13:         **end for**
14:         $c$ sends $\{s_{c,i}^{(t)} \mathbf{H}_{c,i}^{(t)}\}_{i \in \mathcal{H}_c^{(t)}}$ to server
15:     **end for**
16:     **for** $i = 1$ **to** $h$ **do**
17:         Update $\mathbf{H}_i^{(t+1)} \leftarrow \dfrac{1}{|\mathcal{C}_i^{(t)}|} \sum_{c \in \mathcal{C}_i^{(t)}} s_{c,i}^{(t)} \mathbf{H}_{c,i}^{(t)}$
18:     **end for**
19: **end for**

---

### 3.1 Parameter Initialization

Fine-tuning efficiency hinges on the initialization of the LoRA parameters. The standard LoRA initialization sets $\mathbf{B} = \mathbf{0}$ and draws $\mathbf{A} \sim \mathcal{N}(0, \sigma^2)$. In RAVAN, this initialization cannot be used since each $\mathbf{B}_i \mathbf{H}_i \mathbf{A}_i$ would remain $\mathbf{0}$ throughout training as $\mathbf{B}_i$ and $\mathbf{A}_i$ are frozen at initialization. Therefore, we must draw non-zero $\mathbf{B}_i$ and $\mathbf{A}_i$ and set $\mathbf{H}_i = \mathbf{0}$ so that fine-tuning starts from the original pretrained weights but updates the LoRA parameters throughout the training procedure. In this section, we provide methods for effective initializations for the $\mathbf{B}_i$ and $\mathbf{A}_i$ parameters. These initializations do not require performing full-parameter fine-tuning of the original LLM weights such as the initialization presented in Fed-SB. Since each $\mathbf{B}_i$ and $\mathbf{A}_i$ parameter are fixed, the expressive power of the sum $\sum_{i=1}^{h} s_i \mathbf{B}_i \mathbf{H}_i \mathbf{A}_i$ is limited by the subspaces spanned by the column spaces of the $\mathbf{B}_i$ and row spaces of the $\mathbf{A}_i$. We test the following two methods to obtain orthogonal subspaces:

- **Random Normal:** Set $\mathbf{B}_i \sim \mathcal{N}(0, \sigma_B^2)$ and $\mathbf{A}_i \sim \mathcal{N}(0, \sigma_A^2)$. In high-dimensional space, their column and row spaces are orthogonal with high probability.
- **Gram-Schmidt:** For the $\mathbf{B}_i$ parameters, we concatenate the $rh$ columns of random normal initialized $[\mathbf{B}_1, \ldots, \mathbf{B}_h] \in \mathbb{R}^{d \times rh}$ and apply the Gram-Schmidt procedure in the column space.

This yields an orthonormal set $\{\tilde{\boldsymbol{b}}_k\}_{k=1}^{rh}$. The orthonormal set can be sliced back into $h$ blocks of width $r$ to form the $\mathbf{B}_i$. For the $\mathbf{A}_i$ parameters, we can apply the Gram-Schmidt procedure in the row space of the concatenated $\mathbf{A}_i$'s. With this initialization, orthogonality holds deterministically.

We benchmark our initializations against a constant initialization where $\mathbf{B}_i = \mathbf{B}_j$, $\mathbf{A}_i = \mathbf{A}_j \; \forall i, j$. We test an additional more flexible initialization benchmark where $\mathbf{B}_i = \mathbf{M}\mathbf{R}_i$, $\mathbf{A}_i = \mathbf{R}_i\mathbf{N}$ for normally distributed $\mathbf{M} \in \mathbb{R}^{d \times r}$, $\mathbf{N} \in \mathbb{R}^{r \times d}$ and invertible $\mathbf{R}_i \in \mathbb{R}^{r \times r}$ which are different for each head. We refer to this baseline as "shared subspace" since the initialization ensures that the column and row spaces of the $\mathbf{B}_i$ and $\mathbf{A}_i$ parameters are identical. On both vision and language tasks, the random normal and Gram–Schmidt initializations deliver the highest test accuracy, confirming that mutually orthogonal $\mathbf{B}_i$ and $\mathbf{A}_i$ increase the effective rank of the update approximation which translates directly into better downstream performance. Full numbers are reported in Section 4.3.

## 3.2 Head Selection Strategies

RAVAN allows clients to choose subsets of the LoRA heads to fine-tune. This is particularly advantageous in FL where client devices often possess widely varying computational capacities. Suppose we have a participating client set in communication round $t$, $\mathcal{C}^{(t)}$. For client $c \in \mathcal{C}^{(t)}$, we define $\widetilde{\mathbf{H}}_{c,i}^{(t)} = s_{c,i}^{(t)}\mathbf{H}_{c,i}^{(t)} \; \forall i \in [1, \ldots, h]$. At the beginning of each local training step, each client evaluates a scoring function $\rho_{c,i}^{(t)} = \text{score}(\widetilde{\mathbf{H}}_{c,i}^{(t)}, \mathcal{D}_c) \; \forall i \in [1, \ldots, h]$ on its local data $\mathcal{D}_c$. In the following, $\|\cdot\|_F$ is the Frobenius norm of the input matrix. The Frobenius norm of a matrix is calculated as the square root of the sum of the squares of all its entries. We employ the following three scoring functions:

- **Random Scoring:** $\rho_{c,i}^{(t)} \sim \text{Unif}(0, 1)$. Heads receive random scores, so clients form their fine-tuning subset by uniformly sampling heads at random.
- **Weight-Based Scoring:** $\rho_{c,i}^{(t)} = \|\widetilde{\mathbf{H}}_{c,i}^{(t)}\|_F$. Heads whose weights have the largest magnitude are assigned a higher score and deemed more influential.
- **Gradient-Based Scoring:** $\rho_{c,i}^{(t)} = \|\nabla_{\widetilde{\mathbf{H}}_{c,i}^{(t)}} \mathcal{L}_c\|_F$ for a single mini-batch with all other heads frozen. Heads whose gradients have the largest magnitude are deemed more influential.

Client $c$ selects the top $K_c$ heads ranked by $\rho_{c,i}^{(t)}$ and forms the selection set $\mathcal{H}_c^{(t)} = \{i \in [1, \ldots, h] \mid i$ is among the top $K_c$ heads$\}$ where the value of $K_c$ depends on client $c$'s computational constraints. During local fine-tuning, client $c$ only updates heads in $\mathcal{H}_c^{(t)}$. Let $\mathcal{C}_i^{(t)}$ denote the set of clients that fine-tuned head $i$ in communication round $t$. The server performs the update:

$$\mathbf{H}_i^{(t+1)} = \begin{cases} \dfrac{1}{|\mathcal{C}_i^{(t)}|} \displaystyle\sum_{c \in \mathcal{C}_i^{(t)}} \left( \widetilde{\mathbf{H}}_{c,i}^{(t)} \right), & |\mathcal{C}_i^{(t)}| > 0, \\ \mathbf{H}_i^{(t)}, & |\mathcal{C}_i^{(t)}| = 0. \end{cases} \tag{8}$$

Equation (8) ensures exact aggregation of the $s_i$ and $\mathbf{H}_i$ parameters by directly averaging their product and reinitializing each $s_i = 1 \; \forall i \in [1, \ldots, h]$ at the start of every communication round. An alternative measure to ensure exact aggregation is to fix each $s_i = 1$ at the start of training. We would then directly average the individual $\mathbf{H}_i$'s so that $\mathbf{H}_i^{(t+1)} = \frac{1}{|\mathcal{C}_i^{(t)}|} \sum_{c \in \mathcal{C}_i^{(t)}} \left( \mathbf{H}_{c,i}^{(t)} \right)$. We compare these two aggregation schemes in Section 4.3 and find consistent improvements with scaling factors.

# 4 Experiments

## 4.1 Experimental Setup

**Dataset and Model Usage.** For image classification, we adopt **ViT-B/16** [11] (85 M parameters) and fine-tune on two benchmarks: (i) CIFAR-100 (50,000 train / 10,000 test images, 100 classes) and (ii) SVHN (73,250 train / 26,032 test digits, 10 classes). For natural-language tasks, we fine-tune **T5-Base** [31] (224 M parameters) on (i) 20 Newsgroups [28] (11,300 train / 7,532 test articles, 20 topics) and (ii) MRQA [14] (516,800 train / 58,221 test examples). The MRQA corpus is the union of six sources (HotpotQA, NaturalQuestions, NewsQA, SearchQA, SQuAD, TriviaQA).

**Federated Partitioning.** We create federated splits with $|\mathcal{C}| = 20$ or $|\mathcal{C}| = 50$ clients. For I.I.D. partitions, clients receive an equal-sized random subsample of the global training set. For non-I.I.D. partitions, we draw client-specific class proportions from a Dirichlet distribution with $\alpha = 0.3$. For MRQA, which lacks class labels, the Dirichlet split is performed over the six constituent sub-datasets.

Table 2: Performance comparison on CIFAR-100 and SVHN.

| | Method | Rank | CIFAR-100 (Acc. %) | | | | SVHN (Acc. %) | | | |
| --- | --- | --- | --- | --- | --- | --- | --- | --- | --- | --- |
| | | | 20 Clients | | 50 Clients | | 20 Clients | | 50 Clients | |
| | | | I.I.D. | Non-I.I.D. | I.I.D. | Non-I.I.D. | I.I.D. | Non-I.I.D. | I.I.D. | Non-I.I.D. |
| | Full-FT | N/A | 89.89 | 86.86 | 89.78 | 85.17 | 95.06 | 90.29 | 94.90 | 89.49 |
| $N_{total} = 1.2$ M | FedIT | 32 | 83.49 | 68.66 | 81.75 | 68.15 | 88.66 | 84.00 | 91.67 | 77.53 |
| | FedEx-LoRA | 32 | 80.56 | 67.45 | 77.82 | 66.58 | 91.94 | 84.30 | 91.51 | 81.63 |
| | FFA-LoRA | 64 | 78.82 | 56.34 | 78.17 | 59.98 | 91.53 | 86.03 | 91.82 | 83.30 |
| | Fed-SB | 221 | 79.27 | 71.48 | 79.06 | 69.51 | 90.94 | 82.25 | 92.74 | 85.30 |
| | RAVAN | 110 | **84.42** | **76.22** | **84.02** | **73.80** | **94.13** | **90.02** | **93.75** | **89.17** |
| $N_{total} = 2.4$ M | FedIT | 64 | 83.82 | 71.01 | 84.04 | 73.23 | 91.39 | 84.68 | 92.06 | 79.31 |
| | FedEx-LoRA | 64 | 79.38 | 50.47 | 79.42 | 57.86 | 91.16 | 74.04 | 92.01 | 74.84 |
| | FFA-LoRA | 128 | 81.39 | 70.31 | 82.13 | 66.81 | 91.95 | 88.06 | 92.07 | 84.24 |
| | Fed-SB | 313 | 83.03 | 73.12 | 83.90 | 71.13 | 92.29 | 86.89 | 92.78 | 82.46 |
| | RAVAN | 156 | **85.04** | **77.20** | **85.55** | **77.81** | **93.92** | **89.41** | **94.28** | **84.34** |

Table 3: Performance comparison on 20 Newsgroups and MRQA.

| | Method | Rank | 20 Newsgroups (Acc. %) | | | | MRQA (F1 %) | | | |
| --- | --- | --- | --- | --- | --- | --- | --- | --- | --- | --- |
| | | | 20 Clients | | 50 Clients | | 20 Clients | | 50 Clients | |
| | | | I.I.D. | Non-I.I.D. | I.I.D. | Non-I.I.D. | I.I.D. | Non-I.I.D. | I.I.D. | Non-I.I.D. |
| | Full-FT | N/A | 71.34 | 69.29 | 71.71 | 70.13 | 62.19 | 62.25 | 62.41 | 62.51 |
| $N_{total} = 2.4$ M | FedIT | 32 | **69.07** | 61.98 | 67.99 | 60.67 | 61.00 | 60.57 | 61.24 | 60.52 |
| | FedEx-LoRA | 32 | 69.04 | 62.52 | **68.19** | 63.33 | 60.99 | **60.68** | **61.40** | 60.56 |
| | FFA-LoRA | 64 | 68.11 | 62.36 | 68.00 | 64.86 | 60.31 | 60.40 | 61.21 | 60.14 |
| | Fed-SB | 221 | 67.15 | 63.10 | 66.69 | 63.98 | 59.93 | 59.73 | 59.96 | 60.01 |
| | RAVAN | 110 | 68.96 | **65.73** | 68.18 | **65.67** | **61.18** | 60.45 | 61.33 | **61.53** |
| $N_{total} = 4.7$ M | FedIT | 64 | **69.36** | 64.41 | 68.12 | 62.67 | 61.25 | 60.75 | 61.39 | 60.26 |
| | FedEx-LoRA | 64 | 68.59 | 65.11 | 67.75 | 64.31 | 61.23 | 60.36 | 61.43 | 60.06 |
| | FFA-LoRA | 128 | 69.33 | 66.22 | 68.42 | 64.86 | 61.50 | 60.50 | 61.66 | 60.12 |
| | Fed-SB | 313 | 68.07 | 64.18 | 67.58 | 65.59 | 60.22 | 60.11 | 60.28 | 60.60 |
| | RAVAN | 156 | 69.29 | **66.45** | **68.89** | **66.85** | **61.82** | **61.33** | **61.73** | **61.26** |

## 4.2 Main Results: Vision and Language

We consider an FL setting with partial client participation where, in each communication round, the server samples three clients uniformly at random. Every selected client performs 50 local training iterations before uploading its update. Note, we intentionally train for 50 mini-batches and not 50 entire traversals of the client's training dataset so that each client performs exactly the same number of forward-backward passes. We evaluate two separate trainable parameter budgets. The upper half of Tables 2 and 3 correspond to the lower budget and the lower half to the higher budget. The RAVAN configuration uses 4 heads where each head $\mathbf{H}_i \in \mathbb{R}^{r \times r}$ and $r$ is the specified rank. All results displayed in the following sections are the averages across 3 random seeds.

RAVAN achieves the best performance among all PEFT methods in all vision configurations and in 11/16 of the language configurations. A key finding is that its advantage widens systematically in the statistically heterogeneous regime. For CIFAR-100 with 50 non-I.I.D. clients, RAVAN exceeds the performance of FedEx-LoRA by 7.2% and FedIT by 5.6% at the lower budget, whereas the corresponding I.I.D. gains are 6.2% and 2.3% respectively. The language results also display larger improvements in the non-I.I.D. paradigm. On 20 Newsgroups, the gap over Fed-SB reaches 2.6% with 20 non-I.I.D. clients in the lower parameter budget. On MRQA, the pretrained T5-Base already attains a strong F1 score, so all PEFT methods yield only modest absolute gains. Consequently, the scores of different approaches are tightly clustered in the lower-budget rows. Even in this regime, in the higher parameter budget setting, RAVAN outperforms every baseline in every MRQA configuration.

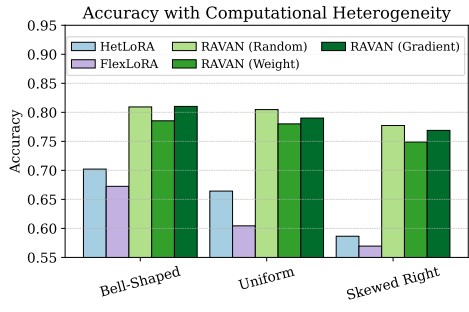

(a) CIFAR-100 Performance Comparison

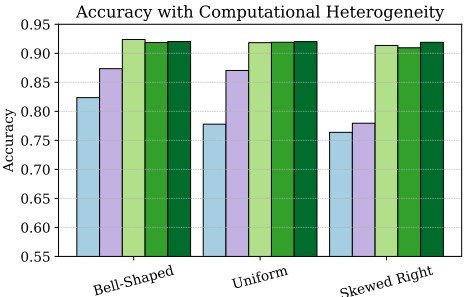

(b) SVHN Performance Comparison

Figure 3: Clients draw trainable parameter budget from bell-shaped, uniform, or skewed right distributions. All RAVAN variants outperform the baselines in every distribution.

## 4.3 Ablation Studies and Analysis

**Initialization Comparison.** Table 4 compares the initializations for the fixed bases $\mathbf{B}_i$, $\mathbf{A}_i$ as described in Section 3.1. The fully orthogonal Gram-Schmidt initialization is consistently best on the vision tasks, adding 1.2% over random normal on SVHN and outperforming the constant baseline by 20% on CIFAR-100. On language tasks, the random normal initialization outperforms the constant baseline by 8.9% on 20 Newsgroups and the shared subspace baseline by 0.26% on MRQA. Thus, the proposed orthogonal initializations maximize the effective rank of the update and yield the strongest accuracy across all domains. While Gram-Schmidt outperforms other schemes on the vision tasks, the procedure is more computationally expensive in high-dimensional space. However, since this initialization is a one-time server-run operation at the start of training, amortized across the entire FL procedure, it adds virtually no overhead to the fine-tuning workload.

Table 4: Initialization comparison with 20 non-I.I.D. clients and lower parameter budget.

| Method | CIFAR-100 | SVHN |
|---|---|---|
| Random Normal | 76.22 | 90.02 |
| Gram-Schmidt | **78.75** | **91.25** |
| Constant | 58.12 | 88.01 |
| Shared Subspace | 57.39 | 84.54 |

| Method | 20 Newsgroups | MRQA |
|---|---|---|
| Random Normal | **65.73** | **60.45** |
| Gram-Schmidt | 64.83 | 59.71 |
| Constant | 56.74 | 60.43 |
| Shared Subspace | 55.64 | 60.19 |

**Computational Heterogeneity.** We emulate devices with unequal trainable parameter budgets by drawing each client's trainable parameter budget from three fixed distributions (bell-shaped, uniform, and skewed right). Details on the individual distributions can be found in the Appendix. As displayed in Figure 3, in these settings, all RAVAN variants outperform the rank-adaptive baselines HetLoRA and FlexLoRA on both CIFAR-100 and SVHN. On CIFAR-100, the strongest baseline loses 11% of overall accuracy when moving from the bell-shaped distribution to the skewed right distribution. In comparison, RAVAN only loses 2% on average across its 3 variants. A key reason is that FlexLoRA performs a per-client SVD and HetLoRA performs hard-rank truncation when redistributing the global model to individual clients. RAVAN avoids these approximation operations, so its updates remain accurate even in the extreme skewed right case. We note, however, that weight-based scoring consistently lags behind the other two scoring mechanisms because it always selects the same high-magnitude heads across all clients. Random scoring and gradient-based scoring more evenly distribute updates across all heads and are better suited for heterogeneous fine-tuning in FL.

Table 5: Effect of using trainable scaling factors with non-I.I.D. clients and lower parameter budget.

| Method | CIFAR-100 | | SVHN | | 20 Newsgroups | | MRQA | |
|---|---|---|---|---|---|---|---|---|
| | 20 Clients | 50 Clients | 20 Clients | 50 Clients | 20 Clients | 50 Clients | 20 Clients | 50 Clients |
| Constant | 74.93 | 71.53 | **90.35** | **89.58** | 65.24 | 65.39 | 60.45 | 61.44 |
| Trainable | **76.22** | **73.80** | 90.02 | 89.17 | **65.73** | **65.67** | **60.60** | **61.53** |

**Influence of Scaling Factors.** Table 5 compares the setting where scaling factors are set to a constant $s_i = 1 \ \forall i \in [1, \ldots, h]$ throughout the fine-tuning procedure in comparison to the standard RAVAN algorithm where the scaling factors are trainable. Keeping the scaling factors trainable boosts

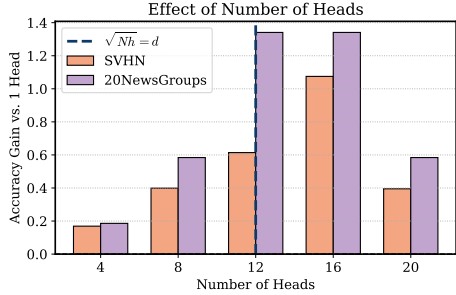
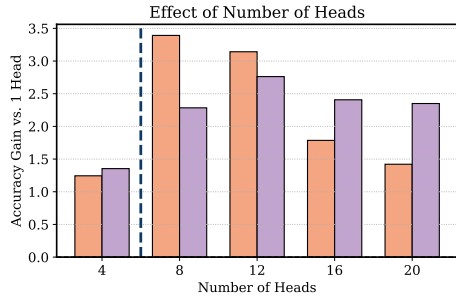

| (a) Lower Parameter Budget Comparison | (b) Higher Parameter Budget Comparison |
|---|---|

Figure 4: Comparison of performance when using different numbers of RAVAN heads at two different parameter budgets (SVHN: $N_{\text{total}}$ = 1.2 M/2.4 M, 20 Newsgroups: $N_{\text{total}}$ = 2.4 M/4.7 M).

accuracy on three of the four datasets while never hurting performance by more than 0.4%. The improvement is largest when client updates are most diverse, specifically in CIFAR-100, as evidenced by Table 2. These lightweight scaling factors upweight useful heads and diminish the importance of heads whose fine-tuning subspaces provide less utility. Notably, we can achieve these performance gains without increasing the per-round communication cost or breaking exact aggregation.

**Effect of Number of Heads.** From Section 2, we know that with a per-layer trainable parameter budget of $N$, the effective rank of the RAVAN update is $\sqrt{Nh}$. However, the maximum effective rank is still bounded by $d$, where the pretrained weights $\mathbf{W} \in \mathbb{R}^{d \times d}$. Hence, adding heads improves the effective rank only while the value of $\sqrt{Nh}$ increases and remains below the dimension $d$. Figure 4 confirms this behavior as accuracy generally only improves while $\sqrt{Nh} < d$. In the lower parameter budget setting, this happens with a larger number of heads as $h$ can assume a larger value while still meeting this condition. In the higher parameter budget setting, we reach this saturation point much sooner. After $\sqrt{Nh}$ exceeds the value of $d$, the effective rank of each individual head becomes smaller while the actual overall update does not become more expressive. This reduces representational power, especially at much larger values of $h$, and weakens performance. Thus, optimally choosing the number of heads is a critical criterion for effective federated fine-tuning using RAVAN.

Table 6: Accuracy comparison on GLUE benchmark with **LLaMA3.2-1B**.

| Method | **MNLI-MM** | **MNLI-M** | **QNLI** | **QQP** | **SST-2** | **RTE** | **Average** |
|---|---|---|---|---|---|---|---|
| FedIT | 84.24 | 84.62 | 82.74 | 85.96 | 94.61 | 65.70 | 82.97 |
| FedEx-LoRA | 84.15 | 84.70 | 82.74 | 86.07 | 94.61 | 65.34 | 82.94 |
| FFA-LoRA | 85.05 | **85.78** | 82.07 | 84.40 | 94.38 | 62.46 | 82.36 |
| Fed-SB | 84.88 | 85.23 | 82.84 | 84.23 | 94.95 | **67.15** | 83.21 |
| RAVAN | **85.24** | 85.65 | **84.00** | **86.11** | **95.18** | **67.15** | **83.90** |

**Scaling to Larger Model Architectures.** We demonstrate the scalability of RAVAN for larger model architectures by benchmarking the method against prior baselines on the GLUE benchmark [37] using **LLaMA3.2-1B** [13] (see Table 6). For each subtask, we use $\mathcal{C} = 20$ clients and follow the training procedure described in Section 4.2. We use a trainable parameter budget of $N_{\text{total}} = 15$ M, which corresponds to the rank configurations in the upper half of Tables 2 and 3. On average, RAVAN outperforms the next best baseline by 0.7% with the maximum gain on a single subtask being the 1.2% improvement over Fed-SB on the QNLI dataset. This demonstrates that RAVAN scales smoothly from 85 M to billion-parameter LLMs and complements state-of-the-art on-device models.

## 5 Conclusion and Future Work

RAVAN offers a new avenue for performing FL fine-tuning in the presence of data and computational heterogeneity. By using multiple augmented LoRA heads and per-head scaling factors, RAVAN

improves the rank of the update approximation within a parameter budget, allowing the method to better approximate the full-parameter update without exceeding a device's memory constraints. Partially freezing subsets of the heads allows clients to adaptively manage their own computational restrictions without sacrificing the accuracy of the aggregated model update. We believe that RAVAN and similar methods will open the doors for LLMs to capitalize on the vast amounts of edge data, a virtually untapped resource for model training and a critical future direction for the frontier of ML.

While RAVAN outperforms existing PEFT benchmarks in FL, we identify three limitations and potential directions for improvement in the current method. First, while clients can fine-tune subsets of the heads to address device-level constraints and computational heterogeneity, the current framework necessitates that the same number of heads be selected in each layer of the original model. This reduces flexibility and can be remedied by a cross-layer scoring scheme that considers all RAVAN heads simultaneously. Second, RAVAN has yet to be tested in the context of differentially-private (DP) learning, and further study is required to validate its performance with stricter privacy guarantees. Finally, data-aware initializations of the $\mathbf{B}_i$ and $\mathbf{A}_i$ parameters may improve the performance of the method while retaining the current improvements in the rank of the update approximation.

## 6 Acknowledgements

This work was supported in part by NSF grants CCF 2045694, CNS-2112471, CPS-2111751, SHF-2107024, ONR grant N00014-23-1-2149, a Google Research Scholar Award, the CyLab IoT Enterprise Security Initiative, and the CMU Prabhu and Poonam Goel fellowship. This work used Bridges-2 GPU at the Pittsburgh Supercomputing Center through allocation CIS250011 from the Advanced Cyberinfrastructure Coordination Ecosystem: Services & Support (ACCESS) program, which is supported by NSF grants #2138259, #2138286, #2138307, #2137603, and #2138296 [7]. We would like to thank Pranay Sharma, Siddharth Shah, Kevin Kuo, Aneesha Sampath, and Akash Dhasade for providing feedback and contributing to discussions for the project.

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

# A Technical Appendices and Supplementary Material

## A.1 Related Works

**Federated Learning.** Federated learning (FL) enables a distributed set of clients to collaboratively train a single global model using their local data [27]. The primary challenges in FL are data heterogeneity and computational heterogeneity induced by the statistical and hardware differences across clients, respectively. Several works aim to address these challenges by altering either the client training procedure or the server aggregation algorithm. FedProx [24], SCAFFOLD [20], and FedDyn [1] add a corrective regularization term to each client's local objective—whether a proximal penalty (FedProx), control-variate correction (SCAFFOLD), or dynamic regularizer (FedDyn)—to keep local updates from drifting too far from the global model, a phenomena more often referred to as "client drift". An alternative approach is to adjust the server aggregation scheme to reduce client drift. For example, FedVARP [49] incorporates historical client updates into the current round's aggregation step to reduce the variance caused by partial client participation and heterogeneity. FedExP [50] varies the server learning rate by using an extrapolation rule that increases the server step size when consecutive aggregated updates point in similar directions and shrinks it when they diverge.

**Parameter-Efficient Fine-Tuning.** Recently, many works have adopted the *pretrain-then-fine-tune* framework, in which a general-purpose large language model (LLM) is adapted to a smaller downstream task [11, 31, 46, 48]. The excessive computational cost of fine-tuning LLMs has led to parameter-efficient fine-tuning (PEFT) methods that fine-tune a fraction of the overall parameters of the model. Adapter tuning [47], BitFit [43], and low-rank adaptation (LoRA) [16] have emerged as effective PEFT methods that can significantly reduce the number of parameters while maintaining task performance. Since its conception, many works have improved upon the initial LoRA formulation in various ways. Works such as QLoRA [45] and LoftQ [51] quantize the pretrained model weights to further improve the memory efficiency of LoRA-based fine-tuning. LoRA-XS [5], LoRA-SB [52], and MoRA [19] introduce an additional LoRA parameter while freezing *both* the model backbone and the original LoRA parameters during fine-tuning. HydraLoRA [35], LoRAMoE [12], and MoLE [53] employ a mixture-of-experts architecture to traditional LoRA-based fine-tuning frameworks.

**PEFT Methods for FL.** As newer applications look to use on-device data to fine-tune LLMs, PEFT methods for FL have become increasingly relevant. FedPETuning [54], FedPrompt [55], and FedIT [41] incorporate adapter tuning, prompt tuning, and LoRA into federated frameworks, respectively. More recently, methods like FFA-LoRA [34], FedEx-LoRA [32], Fed-SB [33], and RoLoRA [44] optimize LoRA in homogeneous-compute FL by addressing the inexactness in LoRA aggregation caused by separately averaging the **B** and **A** parameters. An orthogonal direction is explored by works such as HetLoRA [9], FlexLoRA [4], and FLoRA [38] that address computational heterogeneity in LoRA-based FL fine-tuning by allowing clients to train LoRA parameters with different ranks. However, an examination of cross-device fine-tuning remains limited in this context as methods like FLoRA have communication costs that scale linearly with the number of clients and communication rounds, making fine-tuning particularly difficult in large-scale FL systems. The goal of RAVAN is to design a PEFT method for FL that addresses data and computational heterogeneity, while scaling effectively to cross-device settings with a large number of clients and communication rounds. In this way, we overcome shortcomings in prior works and enable resource-aware fine-tuning.

## A.2 Broader Impacts

RAVAN enables accurate, efficient fine-tuning in a federated setting. While RAVAN has not yet been integrated into a real-world FL system, we identify two potential impacts of utilizing RAVAN:

- **Edge Data for LLM Fine-Tuning:** RAVAN provides an opportunity for LLMs to utilize the data collected by edge devices. As edge applications become increasingly critical, capitalizing on these specialized datasets can make these applications more effective without forfeiting data locality.
- **Efficiency and Improved Performance:** LLM fine-tuning is an expensive, energy-consuming procedure. RAVAN improves the efficiency of the process while retaining performance. Applied to realistic FL training regimes, RAVAN addresses some of these prior concerns.

RAVAN should be implemented with safeguards to prevent misuse, privacy leaks, and harmful content. While RAVAN does not exacerbate these issues, they remain concerns in LLM usage more broadly.

### A.3 Experimental Settings

**Hyperparameters and Optimization Details.** In this section, we highlight hyperparameter choices used in our experiments that were not discussed in Section 4. These descriptions, in addition to the provided code, should aid in the reproducibility of the stated results.

Table 7: FL hyperparameter settings used for each model–dataset pair.

|  | Model/Dataset | | | | |
| --- | --- | --- | --- | --- | --- |
|  | ViT-B-16/CIFAR-100 | ViT-B-16/SVHN | T5-Base/20 Newsgroups | T5-Base/MRQA | LLaMA3.2-1B/GLUE |
| Batch Size | 32 | 32 | 32 | 32 | 16 |
| Max Sequence Length | – | – | 256 | 256 | 128 |
| Local Iterations | 50 | 50 | 50 | 50 | 50 |
| Communication Rounds | 50 | 50 | 100 | 20 | 20 |
| Total Epochs (per round) | 1 | 1 | 1 | 1 | 1 |

For RAVAN and each baseline, we run a learning rate hyperparameter sweep across the values $\{5e-5, 1e-5, 5e-4, 1e-4, 5e-3, 1e-3, 5e-2, 1e-2, 5e-2\}$ and choose the most performant learning to represent in our results. Table 8 represents the optimal choices for each baseline in all settings. The following results each use the ADAM optimizer with momentum set to $0.9$.

Table 8: Optimal learning rate configurations for all baselines.

(a) Lower parameter budget / I.I.D. clients.

| Method | Model/Dataset | | | | |
| --- | --- | --- | --- | --- | --- |
|  | ViT-B-16/CIFAR-100 | ViT-B-16/SVHN | T5-Base/20 Newsgroups | T5-Base/MRQA | LLaMA3.2-1B/GLUE |
| FedIT | $5\times10^{-3}$ | $1\times10^{-3}$ | $1\times10^{-3}$ | $1\times10^{-3}$ | $1\times10^{-4}$ |
| FedEx-LoRA | $1\times10^{-3}$ | $1\times10^{-3}$ | $1\times10^{-3}$ | $1\times10^{-3}$ | $1\times10^{-4}$ |
| FFA-LoRA | $1\times10^{-2}$ | $1\times10^{-2}$ | $1\times10^{-2}$ | $5\times10^{-3}$ | $1\times10^{-3}$ |
| Fed-SB | $1\times10^{-3}$ | $5\times10^{-3}$ | $5\times10^{-3}$ | $1\times10^{-3}$ | $1\times10^{-4}$ |
| RAVAN | $5\times10^{-4}$ | $5\times10^{-4}$ | $5\times10^{-4}$ | $1\times10^{-4}$ | $5\times10^{-5}$ |

(b) Higher parameter budget / I.I.D. clients.

| Method | Model/Dataset | | | | |
| --- | --- | --- | --- | --- | --- |
|  | ViT-B-16/CIFAR-100 | ViT-B-16/SVHN | T5-Base/20 Newsgroups | T5-Base/MRQA | LLaMA3.2-1B/GLUE |
| FedIT | $5\times10^{-3}$ | $1\times10^{-3}$ | $1\times10^{-3}$ | $5\times10^{-4}$ | $1\times10^{-4}$ |
| FedEx-LoRA | $1\times10^{-3}$ | $1\times10^{-3}$ | $1\times10^{-3}$ | $5\times10^{-4}$ | $1\times10^{-4}$ |
| FFA-LoRA | $1\times10^{-2}$ | $1\times10^{-2}$ | $1\times10^{-2}$ | $1\times10^{-2}$ | $1\times10^{-3}$ |
| Fed-SB | $1\times10^{-3}$ | $1\times10^{-3}$ | $5\times10^{-3}$ | $5\times10^{-4}$ | $1\times10^{-4}$ |
| RAVAN | $5\times10^{-4}$ | $5\times10^{-4}$ | $1\times10^{-4}$ | $1\times10^{-4}$ | $5\times10^{-5}$ |

(c) Lower parameter budget / non-I.I.D. clients.

| Method | Model/Dataset | | | |
| --- | --- | --- | --- | --- |
|  | ViT-B-16/CIFAR-100 | ViT-B-16/SVHN | T5-Base/20 Newsgroups | T5-Base/MRQA |
| FedIT | $5\times10^{-3}$ | $1\times10^{-3}$ | $1\times10^{-3}$ | $1\times10^{-3}$ |
| FedEx-LoRA | $1\times10^{-3}$ | $1\times10^{-3}$ | $1\times10^{-3}$ | $1\times10^{-3}$ |
| FFA-LoRA | $1\times10^{-2}$ | $1\times10^{-2}$ | $1\times10^{-2}$ | $5\times10^{-3}$ |
| Fed-SB | $5\times10^{-4}$ | $5\times10^{-3}$ | $1\times10^{-3}$ | $5\times10^{-4}$ |
| RAVAN | $5\times10^{-4}$ | $5\times10^{-4}$ | $5\times10^{-4}$ | $1\times10^{-4}$ |

(d) Higher parameter budget / non-I.I.D. clients.

| Method | Model/Dataset | | | |
| --- | --- | --- | --- | --- |
|  | ViT-B-16/CIFAR-100 | ViT-B-16/SVHN | T5-Base/20 Newsgroups | T5-Base/MRQA |
| FedIT | $5\times10^{-3}$ | $1\times10^{-3}$ | $1\times10^{-3}$ | $5\times10^{-4}$ |
| FedEx-LoRA | $1\times10^{-3}$ | $5\times10^{-4}$ | $1\times10^{-3}$ | $5\times10^{-4}$ |
| FFA-LoRA | $1\times10^{-2}$ | $1\times10^{-2}$ | $1\times10^{-2}$ | $5\times10^{-3}$ |
| Fed-SB | $5\times10^{-4}$ | $1\times10^{-3}$ | $1\times10^{-3}$ | $5\times10^{-4}$ |
| RAVAN | $5\times10^{-4}$ | $5\times10^{-4}$ | $1\times10^{-4}$ | $1\times10^{-4}$ |

**Baseline Descriptions.** We provide details for each of the baselines used in our experiments. We highlight how each baseline initializes, trains, and communicates the individual LoRA parameters.

- **FedIT:** FedIT initializes the LoRA parameters with $\mathbf{B}^{(0)} = \mathbf{0}$ and $\mathbf{A}^{(0)} \sim \mathcal{N}(0, \sigma^2)$. In communication round $t$, each client $c \in \mathcal{C}^{(t)}$ locally trains the LoRA parameters resulting in local parameters $\mathbf{B}_c^{(t)}$, $\mathbf{A}_c^{(t)}$. After communicating these parameters back to the central server, the server performs the following aggregation to generate the new global model:

$$\mathbf{B}^{(t+1)} = \frac{1}{|\mathcal{C}^{(t)}|} \sum_{c \in \mathcal{C}^{(t)}} \mathbf{B}_c^{(t)}, \quad \mathbf{A}^{(t+1)} = \frac{1}{|\mathcal{C}^{(t)}|} \sum_{c \in \mathcal{C}^{(t)}} \mathbf{A}_c^{(t)} \tag{9}$$

- **FedEx-LoRA:** FedEx-LoRA initializes the LoRA parameters with $\mathbf{B}^{(0)} = \mathbf{0}$ and $\mathbf{A}^{(0)} \sim \mathcal{N}(0, \sigma^2)$. In communication round $t$, each client $c \in \mathcal{C}^{(t)}$ locally trains the LoRA parameters resulting in local parameters $\mathbf{B}_c^{(t)}$, $\mathbf{A}_c^{(t)}$. To address the exact aggregation issue, the server updates both the global LoRA parameters as well as the model backbone:

$$\mathbf{B}^{(t+1)} = \frac{1}{|\mathcal{C}^{(t)}|} \sum_{c \in \mathcal{C}^{(t)}} \mathbf{B}_c^{(t)}, \quad \mathbf{A}^{(t+1)} = \frac{1}{|\mathcal{C}^{(t)}|} \sum_{c \in \mathcal{C}^{(t)}} \mathbf{A}_c^{(t)}$$

$$\mathbf{W}^{(t+1)} = \mathbf{W}^{(t)} + \left( \frac{1}{|\mathcal{C}^{(t)}|} \sum_{c \in \mathcal{C}^{(t)}} \mathbf{B}_c^{(t)} \mathbf{A}_c^{(t)} - \frac{1}{|\mathcal{C}^{(t)}|} \sum_{c \in \mathcal{C}^{(t)}} \mathbf{B}_c^{(t)} \cdot \frac{1}{|\mathcal{C}^{(t)}|} \sum_{c \in \mathcal{C}^{(t)}} \mathbf{A}_c^{(t)} \right) \tag{10}$$

While this ensures exact updates in every round, the updated model backbone $\mathbf{W}^{(t+1)}$ also has to be communicated from the central server, increasing the communication overhead of the procedure.

- **FFA-LoRA:** FFA-LoRA initializes the LoRA parameters with $\mathbf{B}^{(0)} = \mathbf{0}$ and $\mathbf{A} \sim \mathcal{N}(0, \sigma^2)$. However, the $\mathbf{A}$ parameter remains frozen at initialization and is never locally trained by the clients and communicated throughout the procedure. Thus, the only update throughout training is the update to the LoRA $\mathbf{B}$ parameter:

$$\mathbf{B}^{(t+1)} = \frac{1}{|\mathcal{C}^{(t)}|} \sum_{c \in \mathcal{C}^{(t)}} \mathbf{B}_c^{(t)} \tag{11}$$

- **Fed-SB:** Fed-SB uses three LoRA parameters which, for the sake of consistency with prior notation, we call $\mathbf{B} \in \mathbb{R}^{d \times r}$, $\mathbf{H} \in \mathbb{R}^{r \times r}$, $\mathbf{A} \in \mathbb{R}^{r \times d}$. The weight update is reparameterized as $\mathbf{BHA}$. To initialize the LoRA parameters, Fed-SB performs an initial round of full-parameter fine-tuning to obtain a full-parameter weight update $\Delta\mathbf{W}_{\text{full}}$. The weight update is decomposed using SVD to get $\Delta\mathbf{W}_{\text{full}} = \mathbf{U\Sigma V}^\top$. $\mathbf{B}$, $\mathbf{H}$, and $\mathbf{A}$ are then initialized as $\mathbf{B} = \mathbf{U}[:, 1:r]$, $\mathbf{H}^{(0)} = \mathbf{\Sigma}[1:r, 1:r]$, $\mathbf{A} = \mathbf{V}^\top[1:r, :]$. $\mathbf{B}$ and $\mathbf{A}$ are frozen at initialization, so the only update is the following:

$$\mathbf{H}^{(t+1)} = \frac{1}{|\mathcal{C}^{(t)}|} \sum_{c \in \mathcal{C}^{(t)}} \mathbf{H}_c^{(t)} \tag{12}$$

- **FlexLoRA:** FlexLoRA allows each client $c \in \mathcal{C}^{(t)}$ to train LoRA parameters with client-specific ranks $r_c$. To aggregate the LoRA parameters, the server performs SVD on $\frac{1}{|\mathcal{C}^{(t)}|} \sum_{c \in \mathcal{C}^{(t)}} \mathbf{B}_c^{(t)} \mathbf{A}_c^{(t)} = \mathbf{U\Sigma V}^\top$. To redistribute the LoRA parameters back to the clients, the central server sends each client $c \in \mathcal{C}^{(t+1)}$ the following LoRA parameters:

$$\mathbf{B}_c^{(t+1)} = \mathbf{U}[:, 1:r_c]\mathbf{\Sigma}[1:r_c, 1:r_c], \quad \mathbf{A}_c^{(t+1)} = \mathbf{V}^\top[1:r_c, :] \tag{13}$$

- **HetLoRA:** Let $r_{\max}$ be the highest rank supported by any client. Each client pads its local parameters to this common shape (with zeros in the unused columns and rows) before upload, so aggregation is still dimensionally consistent. The server weights the individual LoRA parameters based on their relative Frobenius norms:

$$S_c^{(t)} = \|\mathbf{B}_c^{(t)} \mathbf{A}_c^{(t)}\|_F, \quad p_c^{(t)} = \frac{S_c^{(t)}}{\sum_{c \in \mathcal{C}^{(t)}} S_c^{(t)}}$$

$$\mathbf{B}^{(t+1)} = \sum_{c \in \mathcal{C}^{(t)}} p_c^{(t)} \mathbf{B}_c^{(t)}, \quad \mathbf{A}^{(t+1)} = \sum_{c \in \mathcal{C}^{(t)}} p_c^{(t)} \mathbf{A}_c^{(t)}, \tag{14}$$

The server then truncates the new global LoRA parameters $\mathbf{B}^{(t+1)}$ and $\mathbf{A}^{(t+1)}$ for each client $c \in \mathcal{C}^{(t+1)}$ so that $\mathbf{B}_c^{(t+1)} = \mathbf{B}^{(t+1)}[:, 1:r_c]$ and $\mathbf{A}_c^{(t+1)} = \mathbf{A}^{(t+1)}[1:r_c, :]$.

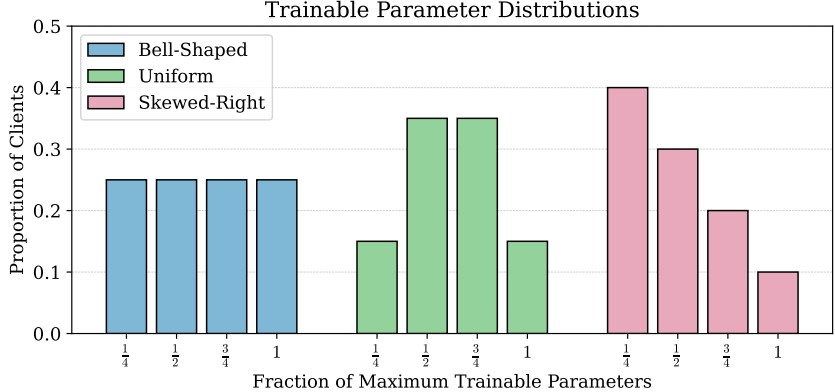

Figure 5: Fraction of clients assigned to each trainable parameter budget in each distribution. Skewed-left is omitted because it is never used in our experiments.

**Computational Heterogeneity Setup.** To emulate computational heterogeneity in our FL setup, we vary the number of trainable parameters at each client. Let $N_{\max}$ denote the largest number of trainable parameters that any client can afford. Each client $c$ is constrained to a trainable parameter budget $N_c \in \{\frac{1}{4}, \frac{1}{2}, \frac{3}{4}, 1\} N_{\max}$. This value is held constant throughout the FL procedure to mirror fixed hardware limits. $N_c$ simply determines the trainable parameter budget per-client; all other hyperparameters are identical to the compute-homogeneous experiments. The bar plot in Figure 5 provides a visual summary of the client mix. Uniform serves as a neutral baseline where there is an equal proportion of clients at every trainable parameter budget; bell-shaped concentrates clients around medium ranks, reflecting the case where most devices have moderate capability; skewed-right stresses the system by placing a large share of the population at the lowest rank, leaving only a small fraction of high-capacity contributors. HetLoRA, FlexLoRA, and RAVAN each accommodate clients with different trainable-parameter budgets in distinct ways:

- **HetLoRA and FlexLoRA:** The LoRA rank for each client $c$ is $r_c = \lfloor (N_c / N_{\max}) \cdot r_{\max} \rfloor$ where $r_{\max}$ is the maximum rank trained by any client. Since the number of trainable parameters scales linearly with the rank, this scaling ensures that every client keeps its update within the allotted budget $N_c$ while allowing higher-capacity devices to contribute proportionally higher-rank updates.
- **RAVAN**: RAVAN uses $H$ LoRA heads per weight matrix. A client with budget $N_c$, fine-tunes only $\lfloor (N_c / N_{\max}) \cdot H \rfloor$ heads and leaves the remaining heads frozen (e.g. for $N_c = \frac{1}{4} N_{\max}$ the client trains one quarter of the heads).

Table 9: Layers equipped with LoRA adapters in each model backbone.

| Model | LoRA Target Modules |
|---|---|
| ViT-B-16 | query, value |
| T5-Base | SelfAttention.q, SelfAttention.v |
| LLaMA3.2-1B | q_proj, v_proj |

**LoRA Implementation Details.** For every model backbone, we insert LoRA adapters only in the self-attention projection matrices. The exact parameters for which we apply LoRA are described in Table 9. All other parameters are frozen and do not have associated LoRA parameters.

**Compute Details and Cluster Description.** All experiments were executed on a GPU cluster managed by SLURM. Each training job used a single NVIDIA V100 32GB GPU with 256 GB RAM. Our environment used Pytorch 2.5.1 and Huggingface 4.47.1 for all experiments. With this setup, each experimental run took $\approx 1$ GPU hour with ViT-B-16 for both CIFAR-100 and SVHN, $\approx 2$ GPU hours with T5-Base for 20 Newsgroups, $\approx 3$ GPU hours with T5-Base for MRQA, and $\approx 2$ GPU hours with LLaMA3.2-1B for each GLUE subtask. All baselines were trained with identical hardware, batch sizes, optimizers, and communication rounds to ensure fair comparison.

## A.4   Additional Experiments

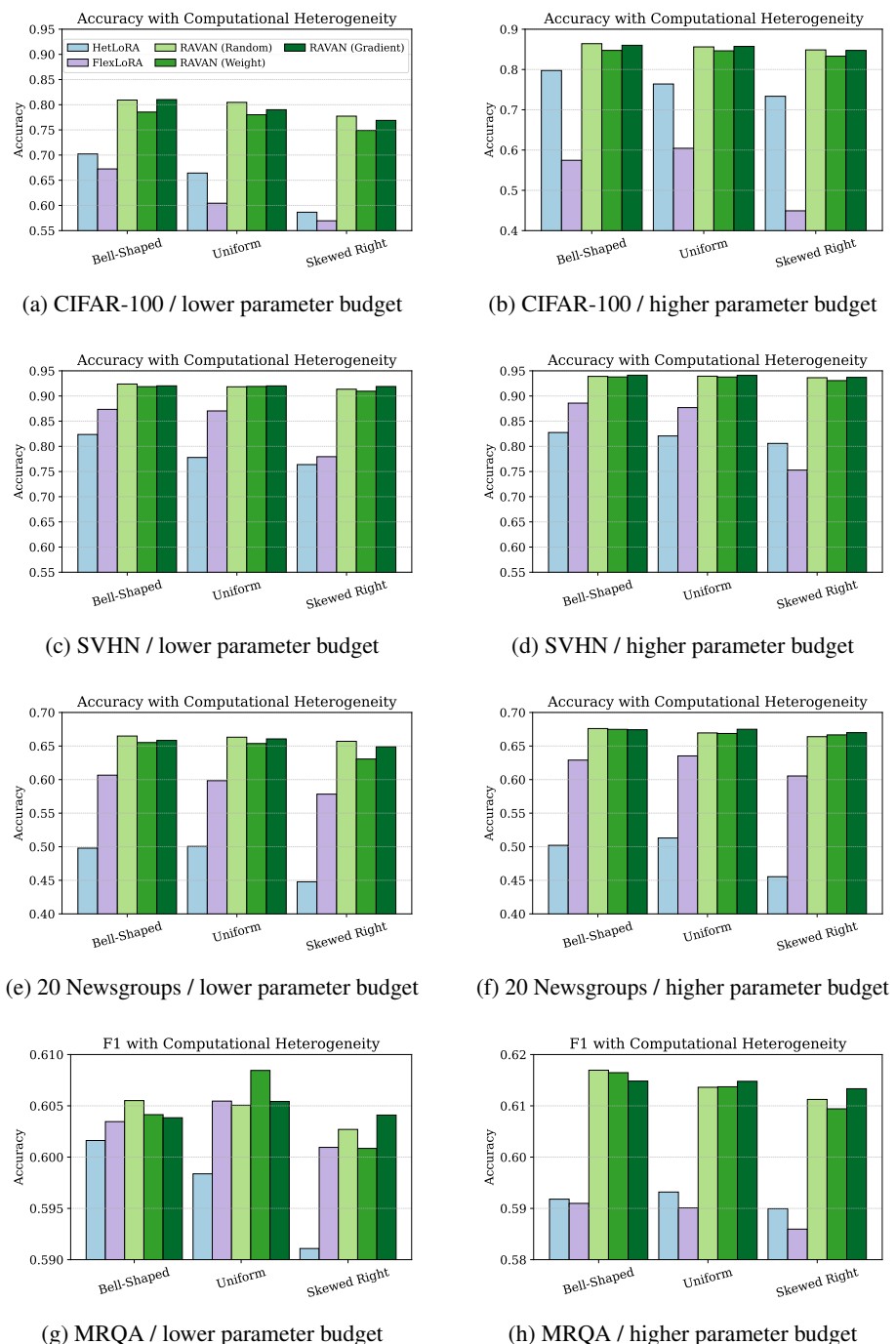

(a) CIFAR-100 / lower parameter budget

(b) CIFAR-100 / higher parameter budget

(c) SVHN / lower parameter budget

(d) SVHN / higher parameter budget

(e) 20 Newsgroups / lower parameter budget

(f) 20 Newsgroups / higher parameter budget

(g) MRQA / lower parameter budget

(h) MRQA / higher parameter budget

Figure 6: Impact of computational heterogeneity on baselines and RAVAN across four datasets. Each row shows a single dataset (left: lower parameter budget, right: higher parameter budget). All settings match the descriptions from Section 4.

**Computational Heterogeneity Experiments.**   We evaluate all methods with 20 non-I.I.D. clients whose usable parameter budgets are drawn from three distributions. Across vision and language tasks, RAVAN's variants consistently outperform the competing LoRA baselines. These results underscore RAVAN's robustness to computational heterogeneity across different tasks and parameter budgets.

Table 10: LoRA ranks under lower vs. higher parameter budgets.

| Method | Lower Budget | Higher Budget |
|---|---|---|
| FedIT | 32 | 64 |
| FedEx-LoRA | 32 | 64 |
| FFA-LoRA | 64 | 128 |
| Fed–SB | 221 | 313 |
| RAVAN | 110 | 156 |

Table 11: Accuracy comparison on GLUE benchmark with LLaMA3.2-1B.

(a) 20 clients / lower parameter budget

| Method | MNLI-MM | MNLI-M | QNLI | QQP | SST-2 | RTE | Average |
|---|---|---|---|---|---|---|---|
| FedIT | 84.24 | 84.62 | 82.74 | 85.96 | 94.61 | 65.70 | 82.97 |
| FedEx-LoRA | 84.15 | 84.70 | 82.74 | 86.07 | 94.61 | 65.34 | 82.94 |
| FFA-LoRA | 85.05 | **85.78** | 82.07 | 84.40 | 94.38 | 62.46 | 82.36 |
| Fed-SB | 84.88 | 85.23 | 82.84 | 84.23 | 94.95 | **67.15** | 83.21 |
| RAVAN | **85.24** | 85.65 | **84.00** | **86.11** | **95.18** | **67.15** | **83.90** |

(b) 20 clients / higher parameter budget

| Method | MNLI-MM | MNLI-M | QNLI | QQP | SST-2 | RTE | Average |
|---|---|---|---|---|---|---|---|
| FedIT | 83.74 | 83.24 | 87.72 | 85.60 | 95.30 | 68.95 | 84.09 |
| FedEx-LoRA | 83.95 | 83.41 | 87.79 | 85.65 | **95.41** | **70.04** | 84.38 |
| FFA-LoRA | 85.27 | 84.69 | 89.51 | 87.10 | 95.18 | 68.23 | 85.00 |
| Fed-SB | 85.85 | 84.76 | 89.53 | 86.09 | 94.95 | 66.79 | 84.66 |
| RAVAN | **86.20** | **85.34** | **90.35** | **87.22** | 95.18 | **70.04** | **85.72** |

(c) 50 clients / lower parameter budget

| Method | MNLI-MM | MNLI-M | QNLI | QQP | SST-2 | RTE | Average |
|---|---|---|---|---|---|---|---|
| FedIT | 84.22 | 84.24 | 87.53 | 85.87 | 94.61 | 61.73 | 83.03 |
| FedEx-LoRA | 84.25 | 84.15 | 87.77 | 85.81 | 94.61 | 62.09 | 83.11 |
| FFA-LoRA | 85.92 | 85.05 | **89.33** | **87.40** | 95.30 | 60.29 | 83.88 |
| Fed-SB | 85.71 | 84.65 | 88.05 | 86.08 | 94.15 | **64.98** | 83.94 |
| RAVAN | **86.03** | **85.53** | 88.91 | 86.95 | **95.41** | 62.09 | **84.15** |

(d) 50 clients / higher parameter budget

| Method | MNLI-MM | MNLI-M | QNLI | QQP | SST-2 | RTE | Average |
|---|---|---|---|---|---|---|---|
| FedIT | 84.66 | 84.26 | 88.38 | 85.87 | 95.18 | 63.58 | 83.66 |
| FedEx-LoRA | 84.74 | 84.02 | 88.50 | 85.82 | 95.41 | 58.12 | 82.77 |
| FFA-LoRA | 85.35 | 84.64 | 87.21 | 87.20 | 94.50 | 61.37 | 83.38 |
| Fed-SB | 85.91 | 85.24 | 87.68 | 86.30 | 93.58 | **67.15** | 84.31 |
| RAVAN | **86.17** | **85.35** | **88.87** | **87.39** | **95.87** | 64.62 | **84.71** |

**LLaMA Experiments.** In these experiments, we use the same hyperparameter settings described in Section 4 but vary the number of total clients and the the ranks of each baseline. Across all four GLUE configurations, RAVAN consistently matches or exceeds the performance of the strongest PEFT baselines using LLaMA3.2-1B (see Table 11). Additionally, while other PEFT baselines vary in performance across settings, RAVAN's performance remains consistent in all configurations. This suggests that RAVAN maintains robust performance, demonstrating its ability to scale effectively to larger models and diverse FL scenarios.

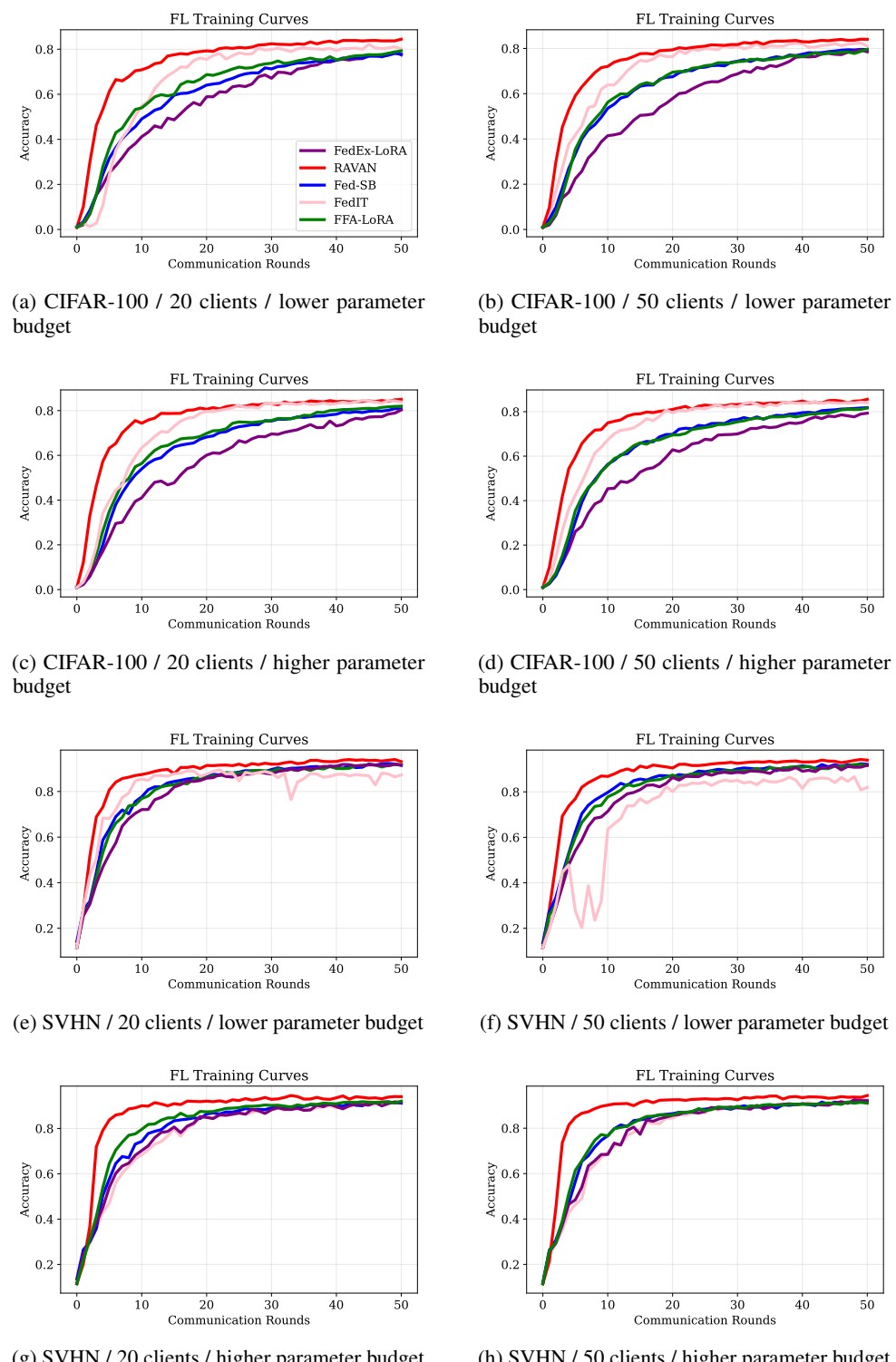

Figure 7: FL training curves for CIFAR-100 and SVHN for all benchmarks.

**Training Curves.** Figure 7 displays the training curves for the various PEFT benchmarks on CIFAR100 and SVHN using a varying number of I.I.D. clients and trainable parameter budgets. In comparison to the other PEFT methods, RAVAN converges faster and to a better overall performance, suggesting that it requires fewer communication rounds to reach optimal performance.

# Appendix References

[43] Elad Ben Zaken, Yoav Goldberg, and Shauli Ravfogel. BitFit: Simple parameter-efficient fine-tuning for transformer-based masked language-models. In Smaranda Muresan, Preslav Nakov, and Aline Villavicencio, editors, *Proceedings of the 60th Annual Meeting of the Association for Computational Linguistics (Volume 2: Short Papers)*, pages 1–9, Dublin, Ireland, May 2022. Association for Computational Linguistics. doi: 10.18653/v1/2022.acl-short.1. URL https://aclanthology.org/2022.acl-short.1/.

[44] Shuangyi Chen, Yue Ju, Hardik Dalal, Zhongwen Zhu, and Ashish J Khisti. Robust federated finetuning of foundation models via alternating minimization of loRA. In *Workshop on Efficient Systems for Foundation Models II @ ICML2024*, 2024. URL https://openreview.net/forum?id=xT0acYbgOF.

[45] Tim Dettmers, Artidoro Pagnoni, Ari Holtzman, and Luke Zettlemoyer. QLoRA: Efficient finetuning of quantized LLMs. In *Thirty-seventh Conference on Neural Information Processing Systems*, 2023. URL https://openreview.net/forum?id=OUIFPHEgJU.

[46] Jacob Devlin, Ming-Wei Chang, Kenton Lee, and Kristina Toutanova. BERT: Pre-training of deep bidirectional transformers for language understanding. In Jill Burstein, Christy Doran, and Thamar Solorio, editors, *Proceedings of the 2019 Conference of the North American Chapter of the Association for Computational Linguistics: Human Language Technologies, Volume 1 (Long and Short Papers)*, pages 4171–4186, Minneapolis, Minnesota, June 2019. Association for Computational Linguistics. doi: 10.18653/v1/N19-1423. URL https://aclanthology.org/N19-1423/.

[47] Neil Houlsby, Andrei Giurgiu, Stanislaw Jastrzebski, Bruna Morrone, Quentin De Laroussilhe, Andrea Gesmundo, Mona Attariyan, and Sylvain Gelly. Parameter-efficient transfer learning for NLP. In Kamalika Chaudhuri and Ruslan Salakhutdinov, editors, *Proceedings of the 36th International Conference on Machine Learning*, volume 97 of *Proceedings of Machine Learning Research*, pages 2790–2799. PMLR, 09–15 Jun 2019. URL https://proceedings.mlr.press/v97/houlsby19a.html.

[48] Jeremy Howard and Sebastian Ruder. Universal language model fine-tuning for text classification. In Iryna Gurevych and Yusuke Miyao, editors, *Proceedings of the 56th Annual Meeting of the Association for Computational Linguistics (Volume 1: Long Papers)*, pages 328–339, Melbourne, Australia, July 2018. Association for Computational Linguistics. doi: 10.18653/v1/P18-1031. URL https://aclanthology.org/P18-1031/.

[49] Divyansh Jhunjhunwala, Pranay Sharma, Aushim Nagarkatti, and Gauri Joshi. Fedvarp: Tackling the variance due to partial client participation in federated learning. In *Uncertainty in Artificial Intelligence*, pages 906–916. PMLR, 2022.

[50] Divyansh Jhunjhunwala, Shiqiang Wang, and Gauri Joshi. Fedexp: Speeding up federated averaging via extrapolation. In *The Eleventh International Conference on Learning Representations*, 2023. URL https://openreview.net/forum?id=IPrzNbddXV.

[51] Yixiao Li, Yifan Yu, Chen Liang, Nikos Karampatziakis, Pengcheng He, Weizhu Chen, and Tuo Zhao. Loftq: LoRA-fine-tuning-aware quantization for large language models. In *The Twelfth International Conference on Learning Representations*, 2024. URL https://openreview.net/forum?id=LzPWWPAdY4.

[52] Kaustubh Ponkshe, Raghav Singhal, Eduard Gorbunov, Alexey Tumanov, Samuel Horvath, and Praneeth Vepakomma. Initialization using update approximation is a silver bullet for extremely efficient low-rank fine-tuning, 2025. URL https://arxiv.org/abs/2411.19557.

[53] Xun Wu, Shaohan Huang, and Furu Wei. Mixture of loRA experts. In *The Twelfth International Conference on Learning Representations*, 2024. URL https://openreview.net/forum?id=uWvKBCYh4S.

[54] Zhuo Zhang, Yuanhang Yang, Yong Dai, Qifan Wang, Yue Yu, Lizhen Qu, and Zenglin Xu. FedPETuning: When federated learning meets the parameter-efficient tuning methods of pretrained language models. In Anna Rogers, Jordan Boyd-Graber, and Naoaki Okazaki, editors, *Findings of the Association for Computational Linguistics: ACL 2023*, pages 9963–9977, Toronto, Canada, July 2023. Association for Computational Linguistics. doi: 10.18653/v1/2023. findings-acl.632. URL `https://aclanthology.org/2023.findings-acl.632/`.

[55] Haodong Zhao, Wei Du, Fangqi Li, Peixuan Li, and Gongshen Liu. Fedprompt: Communication-efficient and privacy-preserving prompt tuning in federated learning. In *ICASSP 2023 - 2023 IEEE International Conference on Acoustics, Speech and Signal Processing (ICASSP)*, pages 1–5, 2023. doi: 10.1109/ICASSP49357.2023.10095356.

