# OpenReview forum: "Ravan: Multi-Head Low-Rank Adaptation for Federated Fine-Tuning"
_NeurIPS.cc/2025/Conference — NeurIPS 2025 poster_

### Official Review · Reviewer_1RFB · 2025-06-20

**Clarity:** 4
**Significance:** 3
**Originality:** 3
**Rating:** 5
**Confidence:** 4

**Summary:**

This paper presents a new approach to federated learning (FL) based on parameter efficient fine-tuning (PEFT) which works with computational and data heterogeneity while allowing for maximum effective rank with low communication overhead and exact aggregation across clients. Specifically, it proposes a multi-head variant of LoRA whereby the weight updates are factored as in LoRA-XS (i.e. \Delta W = B H A) and each client updates a set of H matrices and their associated scalar scaling factors i.e. (\Delta W = \sum_{i=1}^{h}  s_i B_i H_i A_i), and the coefficients A_i and B_i are frozen. For a given parameter budget N, the method allows for a higher effective rank \sqrt(N*h) compared to LoRA (N/2d). Also, it allows for exact aggregation across clients along with communication efficiency. Results on vision and language tasks show that the approach improves performance over other PEFT approaches (e.g. FedIT/Fed-SB). Ablations are shown to compare performance as a function of the mode of initialization of the B/A matrices, head selection strategies, computational heterogeneity, scaling factors, number of heads and scaling.

**Questions:**

* Sec 4.3 / Figure 3: Computational Heterogeneity: Could you provide more details about how the clients choose  # of heads based on their computational capacity? Also, it would be useful to show the distribution of # of heads chosen across clients for each of the 3 distributions: Bell-shaped/Uniform/Skewed right
* Effect of number of heads (Figure 4): It would be good to understand what happens when \sqrt(N*h) exceeds d. Why does the effective rank of each head become smaller in this setup? Could you provide further insight?

**Ethical Concerns:**

["NO or VERY MINOR ethics concerns only"]

**Final Justification:**

The authors' responses have adequately answered my questions - Hence, I am increasing the score.

**Limitations:**

Yes

**Quality:**

3

**Strengths And Weaknesses:**

Strengths:
* Proposes a new PEFT method for FL which is able to tackle computational and data heterogeneity while allowing for a large effective rank.
* Shows that the method outperforms various FL PEFT approaches on vision and language tasks.
* Reports a number of ablations which show the impact of key factors such as initialization, scaling factors
* Approach improves upon prior approaches which either have inexact aggregation (FedIT) or exact aggregation with high communication costs (FFA-LoRA).


Weaknesses:
* Results are reported on simulated data, where the heterogeneity has been introduced artificially (e.g. by drawing client specific class proportions based on a Dirichlet distribution with known parameters).  It is unclear if the method can capture heterogeneity in real datasets. It would be useful to report performance on alternative datasets used in FL research e.g. FedJAX (https://github.com/google/fedjax) contains prepackaged datasets such as EMNIST-62, Stack Overflow grouped by writers/username which would allow experimentation to study data heterogeneity.

---

> ### Author Rebuttal · Authors · 2025-07-30
>
> Thank you for the thorough assessment of our paper and valuable feedback. We hope to comprehensively respond to each of your concerns.
>
> **W1. Results are reported on simulated data, where the heterogeneity has been introduced artificially. It is unclear if the method can capture heterogeneity in real datasets. It would be useful to report performance on alternative datasets used in FL research e.g. FedJAX (https://github.com/google/fedjax) contains prepackaged datasets such as EMNIST-62, Stack Overflow**
>
> Thank you for the suggestion; based on this comment we have included results on federated EMNIST-62 where the dataset has a natural non-I.I.D. partition based on the writer responsible for specific handwritten characters. We use the ViT-B/16 model and the hyperaparameter setup described in the lower halves of Tables 2 and 3 in Section 4 of our paper. We train 3 different settings in which we randomly sample a total of 20 clients, 50 clients, or 100 clients for model fine-tuning and where each client corresponds to a specific writer and their dataset corresponds to their handwritten characters.
>
> FEMNIST-62 with 20 Clients -
> | FedIT | FedEx-LoRA | FFA-LoRA | Fed-SB | Ravan |
> | -------- | -------- | -------- | -------- | -------- |
> | 78.97    | 79.01    | 78.54     | 77.75    | 80.51    |
>
> FEMNIST-62 with 50 Clients -
> | FedIT | FedEx-LoRA | FFA-LoRA | Fed-SB | Ravan |
> | -------- | -------- | -------- | -------- | -------- |
> | 78.96    | 77.99    | 79.37     | 76.99    | 82.05    |
>
> FEMNIST-62 with 100 Clients -
> | FedIT | FedEx-LoRA | FFA-LoRA | Fed-SB | Ravan |
> | -------- | -------- | -------- | -------- | -------- |
> | 81.07    | 80.07    | 79.65     | 79.97    | 81.91    |
>
> As is evident, even when using a dataset with a natural non-I.I.D. partition, Ravan outperforms related federated PEFT methods. Thus, the method is not just applicable in simulated non-I.I.D. datasets but also practically useful in real-world settings. We will include this experiment in our final paper.
>
> **Q1. Could you provide more details about how the clients choose # of heads based on their computational capacity? Also, it would be useful to show the distribution of # of heads chosen across clients for each of the 3 distributions.**
>
> For the sake of space, the discussion on how clients choose the number of heads for fine-tuning was included in the Appendix as opposed to the main text of the paper. The distribution of number of heads chosen across clients for the 3 distributions can be found in Figure 5 of the Appendix. We define $N_{\max}$ as the largest number of trainable parameters that any client can afford. Each client $c$ is constrained to a trainable parameter budget $N_c \in \\{\frac{1}{4}, \frac{1}{2}, \frac{3}{4}, 1\\} N_{\max}$. A client with budget $N_c$, fine-tunes only $\bigl\lfloor \left(N_c \\, \/ \\, N_{\max}\right) \cdot H \bigr\rfloor$ heads and leaves the remaining heads frozen (e.g. for $N_c = \frac{1}{4} N_{\max}$ the client trains one quarter of the heads).
>
> **Q2. It would be good to understand what happens when $\sqrt{N*h}$ exceeds d. Why does the effective rank of each head become smaller in this setup? Could you provide further insight?**
>
> An important thing to note for Figure 4 is that while we increase the number of heads, we are still keeping the number of trainable parameters constant. In order to do so, the size of each individual head must decrease, otherwise the number of trainable parameters would also increase. As a consequence, the effective rank of each individual head goes down. This is fine as long as the effective rank of the overall sum continues to increase. However, beyond a certain number of heads, the effective rank of the overall sum equals the embedding dimension $d$, which is the maximum possible rank of the update approximation, and no longer increases. At this point, as each individual head becomes less expressive, model performance begins to degrade. This phenomenon can be observed in our results figure.

---

> > ### Comment · Reviewer_1RFB · 2025-08-01
> >
> > Thanks for the clarifications.

---

### Official Review · Reviewer_SFDe · 2025-07-02

**Clarity:** 3
**Significance:** 3
**Originality:** 3
**Rating:** 5
**Confidence:** 3

**Summary:**

This paper proposes a contribution to the field of Federated Learning with LLM models. A shortcoming of FL for LLMs is the high communication budget between clients and the server. Previous works in the literature, such as FedIT, proposed using LoRA to reduce this communication cost. This work refines that method, proposing a multi-head LoRA adaptation in the fashion of HydraLoRA. Here, however, the authors propose an augmented version si​Bi​Hi​Ai​ that allows for exact aggregation and increased expressivity. In this version, only si​ and Hi​ are trained and communicated, while Ai​ and Bi​ remain frozen. Experimental results advocate for good efficiency on a diverse set of architectures and datasets.

**Questions:**

- Could the authors provide more grounded explanations for the strategies proposed in parts 3.1 and 3.2? Specifically, why does Gram-Schmidt perform better, and is there a theoretical justification for this choice?

- Can the authors perform new experiments with a more reasonable number of local epochs for client training? This would be necessary to position the results in the context of federated learning and make them comparable with other works in the literature.

- How are the "scaling factors" learned in the proposed method? A clearer explanation of this process would be beneficial for understanding the overall approach.

**Ethical Concerns:**

["NO or VERY MINOR ethics concerns only"]

**Final Justification:**

Authors have addressed my few concerns in the rebuttal. I changed my evaluation to 5: Accept.

**Limitations:**

Yes

**Quality:**

3

**Strengths And Weaknesses:**

Strengths:
- The method is well-grounded and motivated, with a solid overview of related work. It is integrated into the broader context of federated learning and low-rank adaptation.
- The method is evaluated on CIFAR-100, which is a challenging dataset in the context of federated learning.
- The comparison to other methods in the same scenario is well-conducted, and the results are good for the proposed method.
- RAVAN is easy to implement, which is a plus for practical applications.

Weaknesses:

- The claim "To our knowledge, RAVAN is the first federated PEFT technique that simultaneously maintains parameter-efficient computation and communication and addresses both data and computational heterogeneity" is too strong. The idea has been around for a long time, with works like FedPara and others. The authors should be more cautious with such claims.
- Parts 3.1 and 3.2 of the paper raise questions about the underlying assumptions and motivations for the proposed strategies. For part 3.1: why does Gram-Schmidt perform better? Is there a theoretical justification? It would also be interesting to compare these results with a pre-trained initialization on the same domain using a random dataset.
- The choice of 50 local epochs for client training is very problematic for comparing results with other works in the literature. In federated learning, it is common to see clients training for 1, 5, 10, or 20 epochs. Training for 50 epochs is almost equivalent to letting each client train their own model, which undermines the federated learning paradigm. It is absolutely necessary to perform new experiments with a more reasonable number of local epochs.
- The paper mentions that the "scaling factors" are learned, but it is not clear how this learning process occurs. This aspect needs to be clarified for better understanding.

---

> ### Author Rebuttal · Authors · 2025-07-30
>
> Thank you for the detailed critique of our work and the recognition of its value in the field of federated fine-tuning. We aim to thoughtfully address the specific points you have raised.
>
> **W1. The claim "To our knowledge, RAVAN is the first federated PEFT technique that simultaneously maintains parameter-efficient computation and communication and addresses both data and computational heterogeneity" is too strong. The idea has been around for a long time, with works like FedPara and others.**
>
> We appreciate the feedback on the framing of our contributions and will revise this sentence in the final draft of our paper to better reflect the benefits of Ravan in the context of current research. The following will be included in the paper should it be accepted "Extending prior efforts such as FedPara, Ravan introduces an integrated framework that maintains parameter‑efficient computation and communication and demonstrates robustness across diverse data and computational heterogeneity in federated settings."
>
> **W2. For part 3.1: why does Gram-Schmidt perform better? Is there a theoretical justification?**
>
> Section 3.1 discusses various strategies for initializing the Ravan parameters. In particular, we find that the Gram-Schmidt initialization and random normal initialization are effective initializations for the $\textbf{B}_i$ and $\textbf{A}_i$ parameters. The random normal initialization yields orthogonality for the $\textbf{B}_i$ and $\textbf{A}_i$ parameters in expectation, while Gram-Schmidt ensures that the $\textbf{B}_i$ and $\textbf{A}_i$ parameters are orthogonal deterministically. This orthogonality is important because the column space of $\sum  s_i\textbf{B}_i\textbf{H}_i\textbf{A}_i$ is a subspace of the span of the $\textbf{B}_i$'s and the row space of $\sum s_i\textbf{B}_i\textbf{H}_i\textbf{A}_i$ is a subspace of the span of the $\textbf{A}_i$'s. Therefore, if we constrain all the $\textbf{B}_i$'s and $\textbf{A}_i$'s to share the same subspace, the rank of $\sum s_i\textbf{B}_i\textbf{H}_i\textbf{A}_i$ collapses to that of a single head. This is the case for both the constant initialization, where $\textbf{B}_i = \textbf{B}_j$ and $\textbf{A}_i = \textbf{A}_j \ \forall i,j$, and in the shared subspace initialization, where $\textbf{B}_i = \textbf{MR}_i, \textbf{A}_i = \textbf{R}_i\textbf{N}$ for normally distributed $\textbf{M}, \textbf{N}$ and invertible $\textbf{R}_i \ \forall i$. For these initializations, we would not benefit from using multiple heads at all since the effective rank of the update approximation would not actually improve. Section 4.3 demonstrates that initializations that do improve the rank of the update approximation (random normal, Gram-Schmidt) outperform methods that do not improve the rank of the update approximation (constant, shared subspace).
>
>
> Further empirical evidence of the improvement to the rank of the update approximation can be demonstrated by the following experiment. For CIFAR-100, we calculated the rank of the update approximation $\sum s_i\textbf{B}_i\textbf{H}_i\textbf{A}_i$ for all 4 initialization schemes (constant, shared subspace, random normal, Gram-Schmidt) after communication round 1 and communication round 50. For the sake of calculating the rank, any singular value less than $1 \times 10^{-6}$ is considered 0. We follow the same hyperparameter setup in Tables 2 and 3 by using 4 heads of rank 156 each.
>
> After communication round 1 -
>
> | Gram-Schmidt | Random Normal | Constant | Shared Subspace |
> | -------- | -------- | -------- | -------- |
> | 624     | 624     | 156     | 156     |
>
> After communication round 50 -
>
> | Gram-Schmidt | Random Normal | Constant | Shared Subspace |
> | -------- | -------- | -------- | -------- |
> | 624     | 624     | 156     | 156     |
>
> These results establish that Ravan achieves the upper bound theoretical rank of 624 ($4 \times 156$) throughout the training procedure when using initializations that result in orthogonal $\textbf{B}_i$ and $\textbf{A}_i$.
>
> **W3. It would also be interesting to compare these results with a pre-trained initialization on the same domain using a random dataset.**
>
> This is an exciting idea and a potential alternative initialization scheme to the ones we include in the paper. We have benchmarked this initialization in which we pretrain the $\textbf{B}_i$ and $\textbf{A}_i$ parameters prior to the FL fine-tuning procedure. The $\textbf{B}_i$ and $\textbf{A}_i$ parameters are subsequently frozen at this pretrained initialization. The following are results on CIFAR-100 and SVHN in which the $\textbf{B}_i$ and $\textbf{A}_i$ parameters are pretrained on random subsets of CIFAR-100 and SVHN, respectively. The hyperparameter settings follow the ones listed in Table 4.
>
> Results for CIFAR-100 -
>
> | Pretrained | Gram-Schmidt | Random Normal | Constant | Shared Subspace |
> | -------- | -------- | -------- | -------- | -------- |
> | 80.49    | 78.75    | 76.22     | 58.12    | 57.39    |
>
> Results for SVHN -
>
> | Pretrained | Gram-Schmidt | Random Normal | Constant | Shared Subspace |
> | -------- | -------- | -------- | -------- | -------- |
> |  89.53    | 91.25    | 90.02   | 88.01  | 84.54    |
>
> The results of this initialization are mixed with improvements on our proposed initializations in CIFAR-100 but a slight decrease in performance on SVHN (though we admit that testing is limited given the time constraints of the rebuttal period). While this direction is certainly interesting, and a potential direction for future work, we would like to bring up that this initialization would likely require pretraining the $\textbf{B}_i$ and $\textbf{A}_i$ parameters at the server since they are quite a bit larger than the $\textbf{H}_i$ parameters and might be difficult to train given device resource constraints. This would require a public dataset at the server that could be used to pretrain the $\textbf{B}_i$ and $\textbf{A}_i$ parameters. Thus, there are some limitations that might make this initialization more difficult in practice. The initializations that we propose are light-weight, effective, and data-independent. That being said, we do believe that the initialization you propose is an intriguing direction for future study.
>
> **W4. The choice of 50 local epochs for client training is very problematic for comparing results with other works in the literature. In federated learning, it is common to see clients training for 1, 5, 10, or 20 epochs.**
>
> We believe there is a misunderstanding here -- we train for 50 *local iterations* and not 50 *local epochs*. This means that for each client, we only train on 50 mini-batches and *not* 50 entire traversals of the client's training dataset. 50 local iterations, depending on the dataset size and batch size, is approximately 1 local epoch (client dataset size / batch size = local iterations per epoch). We use local iterations instead of local epochs since clients may have different amounts of training data, so using local iterations means that each client performs exactly the same number of forward-backward passes. We believe that this is a fair amount of compute for resource-constrained devices and does not differ from most modern FL literature in a significant way. We will highlight the difference between local iterations and local epochs in the final draft of our paper to avoid similar confusion in the future. However, to demonstrate the robustness of our method, we include the results for CIFAR-100 and SVHN using 1 or 2 local epochs below.
>
> Results for CIFAR-100 with 1 Local Epoch -
> | FedIT | FedEx-LoRA | FFA-LoRA | Fed-SB | Ravan |
> | -------- | -------- | -------- | -------- | -------- |
> | 75.21    | 61.32    | 68.47     | 73.34    | 79.12    |
>
> Results for CIFAR-100 with 2 Local Epochs -
> | FedIT | FedEx-LoRA | FFA-LoRA | Fed-SB | Ravan |
> | -------- | -------- | -------- | -------- | -------- |
> | 72.36    | 77.36    | 74.66     | 77.61    | 80.88    |
>
> Results for SVHN with 1 Local Epoch -
> | FedIT | FedEx-LoRA | FFA-LoRA | Fed-SB | Ravan |
> | -------- | -------- | -------- | -------- | -------- |
> | 92.70    | 91.40    | 92.77     | 92.82    | 93.15    |
>
> Results for SVHN with 2 Local Epochs -
> | FedIT | FedEx-LoRA | FFA-LoRA | Fed-SB | Ravan |
> | -------- | -------- | -------- | -------- | -------- |
> | 93.91    | 92.43    | 93.37     | 94.02    | 94.53    |
>
> **W5. The paper mentions that the "scaling factors" are learned, but it is not clear how this learning process occurs. This aspect needs to be clarified for better understanding.**
>
> Effectively, the scaling factors $s_i$ are trainable parameters just like any other trainable parameter in the network. The only difference is that they are scalars and not multi-dimensional tensors. However, they receive gradients, just like the $\textbf{H}_i$ parameters, and are trained with these gradient updates. The advantage to keeping these $s_i$ factors trainable is that the network naturally learns which heads are more significant throughout the fine-tuning procedure and as a consequence adjusts the value of $s_i$, in essence learning the scaling factors most beneficial for model prediction.

---

### Official Review · Reviewer_pUXf · 2025-07-03

**Clarity:** 4
**Significance:** 3
**Originality:** 3
**Rating:** 4
**Confidence:** 4

**Summary:**

This paper introduces RAVAN, a multi-head, low-rank adaptation method for the federated fine-tuning of large models. The proposed method addresses the challenges of data and computational heterogeneity in federated learning by reparameterizing weight updates as a sum of multiple augmented LoRA heads. By freezing the outer matrices and only training the small inner matrices and lightweight scaling factors, RAVAN aims to improve the expressive power of the model updates while maintaining parameter efficiency and ensuring mathematically exact aggregation.

**Questions:**

While RAVAN offers a high theoretical rank, does the optimization process for non-I.I.D. updates actually utilize this full capacity? I'm interested in an empirical analysis of the singular values of the learned update matrix which demonstrates how much of the available rank is used in practice. This would confirm if the increased rank is a fully leveraged resource in heterogeneous settings or primarily a theoretical upper bound.

**Ethical Concerns:**

["NO or VERY MINOR ethics concerns only"]

**Final Justification:**

The author addresses most of my concerns. I will maintain my positive score.

**Limitations:**

Yes

**Quality:**

3

**Strengths And Weaknesses:**

Strengths:

1. **Exact Aggregation**: RAVAN's design provides an elegant solution to the inexact aggregation problem that affects many federated LoRA methods. By having clients only train and upload $s_iH_i$, the server can perform a direct, weighted average that is mathematically equivalent to averaging the full updates.
2. **Communication and Computation Efficiency**: The method maintains the low communication cost of standard LoRA by only requiring clients to send the small, trained parameters. Furthermore, it directly addresses computational heterogeneity by allowing resource-constrained clients to train only a subset of the available heads, making the framework flexible and scalable in realistic FL environments.

Weaknesses:

One of the paper's core argument is that RAVAN solves the non-I.I.D. issue, based on the observation that the spectrum of weight updates is broader in such settings.  While the empirical results are strong, this connection feels more like an intuition than a direct, proven solution. The observation in Figure 1 that $\delta W$ has a more distributed rank does not necessarily mean that a low-rank approximation is the primary bottleneck for non-I.I.D. training. A more rigorous theoretical justification, perhaps related to convergence guarantees or generalization bounds, would be appreciated to formally connect the spectral properties of non-I.I.D. updates to the necessity of a higher-rank adaptation space.

---

> ### Author Rebuttal · Authors · 2025-07-30
>
> Thank you for the meaningful feedback and the acknowledgement of Ravan's strengths and practical utility. We would like to address your individual concerns.
>
> **W1. A more rigorous theoretical justification would be appreciated to formally connect the spectral properties of non-I.I.D. updates to the necessity of a higher-rank adaptation space.**
>
> Thank you for bringing up this discussion; we believe this is a valid point about the theoretical rigor of our proposed method. Unfortunately, there exists limited theory about the spectral properties of LoRA optimization, especially in distributed or federated settings. As such, our work closely aligns with previous works in federated fine-tuning that focus primarily on the empirical advantages of methodological innovations ([1], [2], [3]). That being said, there are a few prior works ([4], [5], [6]) that tie the convergence of LoRA directly to the rank and suggest that when the actual desired update is high rank, LoRA may underperform or even fail to converge. This is particularly important given our observation that the true update is high rank in non-I.I.D. settings. In this context, we believe there are two critical pieces of evidence that suggest that the rank of the update approximation contributes to the improvement in performance in federated settings.
>
> * Increasing the number of heads, and subsequently the rank of the update approximation, improves the performance of the method *only until* the rank of the update approximation stops increasing. In Section 4.3, we include an ablation that demonstrates the impact of changing the number of heads while leaving the number of trainable parameters constant. We find that increasing the number of heads is beneficial as long as the rank of the update approximation remains less than the embedding dimension, which is the maximum possible rank of the update approximation. However, once the rank of the update approximation equals the embedding dimension and stops increasing, these performance gains vanish. Since we hold the number of trainable parameters constant in this experiment, this conveys that the rank of the update approximation is associated directly with the performance of the method.
> * Initialization schemes that do not increase the effective rank of the update approximation perform worse on all benchmarks. In Section 3.1, we propose two initialization benchmarks that do not increase the effective rank of the update approximation. In the constant initialization $\textbf{B}_i = \textbf{B}_j$ and $\textbf{A}_i = \textbf{A}_j \ \forall i,j$, and in the shared subspace initialization $\textbf{B}_i = \textbf{MR}_i, \textbf{A}_i = \textbf{R}_i\textbf{N}$ for normally distributed $\textbf{M}, \textbf{N}$ and invertible $\textbf{R}_i \ \forall i$. We highlight that the constant initialization and shared subspace initialization underperform relative to the random normal initialization and Gram-Schmidt initialization. Despite using multiple heads, the constant and shared subspace initializations have an update approximation that is the same rank as a single head. In contrast, the random normal initialization and Gram-Schmidt initialization have an effective rank that is (number of heads * rank of each head). Since each initialization uses the same number of trainable parameters, the same rank for each head, and the same number of heads, the only variable that changes is the effective rank of the entire update approximation which qualifies the importance of increasing the rank of the update approximation.
>
> While we agree that our work presents limited theoretical insights into this behavior, we believe these experiments and prior theoretical work combined present convincing evidence that the improved rank of Ravan's update approximation is a critical component of the method's performance.
>
> **Q1. While Ravan offers a high theoretical rank, does the optimization process for non-I.I.D. updates actually utilize this full capacity?**
>
> Yes, the optimization process actually utilizes this full capacity as demonstrated by the following experiment. For CIFAR-100, we calculated the rank of the update approximation $\sum s_i\textbf{B}_i\textbf{H}_i\textbf{A}_i$ for all 4 initialization schemes (constant, shared subspace, random normal, Gram-Schmidt) after communication round 1 and communication round 50. For the sake of calculating the rank, any singular value less than $1 \times 10^{-6}$ is considered 0. We follow the same hyperparameter setup in Tables 2 and 3 by using 4 heads of rank 156 each.
>
> After communication round 1 -
>
> | Gram-Schmidt | Random Normal | Constant | Shared Subspace |
> | -------- | -------- | -------- | -------- |
> | 624     | 624     | 156     | 156     |
>
> After communication round 50 -
>
> | Gram-Schmidt | Random Normal | Constant | Shared Subspace |
> | -------- | -------- | -------- | -------- |
> | 624     | 624     | 156     | 156     |
>
> These results establish that Ravan achieves the upper bound theoretical rank of 624 ($4 \times 156$) throughout the training procedure when using initializations that result in orthogonal $\textbf{B}_i$ and $\textbf{A}_i$. It also lends further credence to the idea that the differentiating factor in performance between the various initialization schemes is a large difference in effective rank. We have included the performance for the different initialization schemes on CIFAR-100 here for convenience.
>
> Results for CIFAR-100 -
>
> | Gram-Schmidt | Random Normal | Constant | Shared Subspace |
> | -------- | -------- | -------- | -------- |
> | 78.75    | 76.22     | 58.12    | 57.39    |
>
> [1] "FedEx-LoRA: Exact Aggregation for Federated and Efficient Fine-Tuning of Foundation Models" Singhal et al.
>
> [2] "FLoRA: Federated Fine-Tuning Large Language Models with Heterogeneous Low-Rank Adaptations" Wang et al.
>
> [3] "Heterogeneous LoRA for Federated Fine-tuning of On-Device Foundation Models" Cho et al.
>
> [4] "Randomized Asymmetric Chain of LoRA: The First Meaningful Theoretical Framework for Low-Rank Adaptation" Malinovsky et al.
>
> [5] "GeLoRA: Geometric Adaptive Ranks for Efficient LoRA Fine-Tuning" Ed-dib et al.
>
> [6] "LoRA Training Provably Converges to a Low-Rank Global Minimum or It Fails Loudly (But it Probably Won't Fail)" Kim et al.

---

### Author Response · Authors · 2025-08-06

Dear Reviewers,

Thank you all for your helpful comments. We would like to recap our discussions with you and summarize the additional experiments we have conducted in this rebuttal phase. The following is a list of the four experiments we have included in our rebuttals in order to address your concerns.

- Rank of each initialization scheme: For the four initializations we include in our paper (constant, shared subspace, random normal, Gram-Schmidt), we calculated the rank of the update approximation $\sum s_i\textbf{B}_i\textbf{H}_i\textbf{A}_i$. We found that the random normal and Gram-Schmidt initializations achieve the maximum possible rank defined by (number of heads * rank of each head) throughout the training procedure. In contrast, the constant and shared subspace initializations have a rank equivalent to that of a single head. This justifies the importance of rank in the optimization procedure and helps explain the difference in performance between these initialization schemes.
- Pretrained initialization: We tested an additional initialization as recommended by Reviewer SFDe in which the $\textbf{B}_i$ and $\textbf{A}_i$ parameters are pretrained on a random subset of the dataset. This initialization shows mixed results with improvements over our proposed initializations on CIFAR-100 and a slight decrease in performance on SVHN. While this is an interesting direction for future work, we would like to point out that our initializations are light-weight, effective initializations that do not require an additional pretraining phase prior to performing fine-tuning.
- Number of epochs: We show that Ravan outperforms other FL PEFT baselines on CIFAR-100 and SVHN when using both 1 and 2 local epochs. This shows that the method is robust to different amounts of local client fine-tuning.
- Natural non-I.I.D. partitions: As explained by Reviewer 1RFB, Ravan was originally only tested on datasets where we artificially generated non-I.I.D. partitions in the data based on the output label of each example. We included a test on federated EMNIST-62 which has a natural non-I.I.D. partition where each client is a writer whose dataset is their set of handwritten digits. In this setting, Ravan still outperforms other baselines. These results are consistent across, 20/50/100 clients.

We hope these experiments have helped clear any concerns about the paper. We would like to ask that, if any reviewer has any lingering comments, to please include them in the Rebuttal threads. If not, it would be particularly helpful to let us know that we have addressed all concerns. Thank you again for the feedback, and we hope to hear from you!

---

### Decision · Program_Chairs · 2025-09-17

**Decision:**

Accept (poster)

**Comment:**

This paper introduces a novel multi-head, low-rank adaptation method for addressing the data and computational heterogeneity challenges in the setting of federated fine-tuning of large models. The effectiveness of the proposed approach is verified by extensive experimental evaluations.  One weakness is the lack of theoretical justifications, also pointed out by one of the the reviewers.

The paper receives all positive scores from the reviewers initially,  and the scores are further increased after rebuttal as the reviewers are satisfied with the authors' rebuttal. Considering the technical novelty and the consensus positive reviews, I would recommend the acceptance of this paper, while encouraging the authors to further improve the manuscript by providing rigorous analysis on theoretical justification of convergence and generalization bounds.